# Multi-agent Markov Entanglement

**Shuze Chen**
Graduate School of Business
Columbia University
New York, NY 10027
`shuze.chen@columbia.edu`

**Tianyi Peng**
Graduate School of Business
Columbia University
New York, NY 10027
`tianyi.peng@columbia.edu`

## Abstract

Value decomposition has long been a fundamental technique in multi-agent reinforcement learning and dynamic programming. Specifically, the value function of a global state $(s_1, s_2, \ldots, s_N)$ is often approximated as the sum of local functions: $V(s_1, s_2, \ldots, s_N) \approx \sum_{i=1}^{N} V_i(s_i)$. This approach has found various applications in modern reinforcement learning systems. However, the theoretical justification for why this decomposition works so effectively remains underexplored. In this paper, we uncover the underlying mathematical structure that enables value decomposition. We demonstrate that a Markov decision process (MDP) permits value decomposition *if and only if* its transition matrix is not "entangled"—a concept analogous to quantum entanglement in quantum physics. Drawing inspiration from how physicists measure quantum entanglement, we introduce how to measure the "Markov entanglement" and show that this measure can be used to bound the decomposition error in general multi-agent MDPs. Using the concept of Markov entanglement, we proved that a widely-used class of policies, the index policy, is weakly-entangled and enjoys a sublinear $\mathcal{O}(\sqrt{N})$ scale of decomposition error for $N$-agent systems. Finally, we show Markov entanglement can be efficiently estimated, guiding practitioners on the feasibility of value decomposition.

## 1 Introduction

Learning the value function given certain policy, or *policy evaluation*, is one of the most fundamental tasks in RL. Significant attention has been paid to single-agent policy evaluation [39, 8, 40]. However, when it comes to multi-agent reinforcement learning (MARL), single-agent methodologies typically suffer from *the curse of dimensionality*: the state space of the system scales exponentially with the number of agents. To tackle this problem, one common technique is value decomposition,

$$V(s_1, s_2, \ldots, s_N) \approx \sum_{i=1}^{N} V_i(s_i),$$

where $V_i$ is some local function that can be learned independently by each agent. It quickly follows that this decomposition greatly reduces the computation complexity from exponential to linear dependency on the number of agents $N$.

The remaining question is whether this decomposition is effective. This is non-trivial due to the coupling of agents—individual agent's action and transition depend on other agents. For example, in a ride-hailing platform, if one driver took the order, then other drivers are not allowed fulfill the same order. As a result, value decomposition may lose information and introduce bias without considering the global constraints.

In the past several decades, both positive and negative results have been reported. Back to the last century, [49, 47] apply Lagrange relaxations to decompose the global value and obtain the well-known

39th Conference on Neural Information Processing Systems (NeurIPS 2025).

Whittle index policy. The Lagrange decomposition idea has also been proved successful in many other important multi-agent tasks such as network revenue management [1, 50], resource allocation [27, 7], and online matching [11, 12, 36, 28]. However, Lagrange decomposition relies on the knowledge of system dynamics and [2] show its decomposition error can be arbitrarily bad for general multi-agent MDPs. In more recent days, practitioners apply online (deep) reinforcement learning to train a local value function for each individual agent. This practice gives birth to state-of-the-art dispatching policies in ride-hailing platforms and has been well recognized by the operations research community, such as DiDi Chuxing [33] (Daniel H. Wagner Prize, 2020) and Lyft [4] (Franz Edelman Laureates, 2024). Intervention policies based on a similar value decomposition idea also demonstrate substantial empirical advantages and have been deployed by a behavioral health platform in Kenya [5] (Pierskalla Award, 2024). In broader MARL literature, value decomposition serves as one key component of centralized training and decentralized execution (CTDE) paradigm, achieving strong empirical performance [38, 29, 35]. However, recent research has started reflecting on the invalidity and potential flaw of value decomposition in practice [25, 16].

Despite all these empirical success and failures, there remains little theoretical understanding of whether and how we can decompose the value function in multi-agent MDPs.

## 1.1 This paper

In this paper, we will uncover the underlying mathematical structure that enables/disables value decomposition. Our new theoretical framework quantifies the inter-dependence of agents in multi-agent MDPs and systematically characterizes the effectiveness of value decomposition. For simplicity, we will demonstrate the main results through two-agent MDPs indexed by agent $A$ and $B$. We later extend our results to general $N$-agent MDPs in Appendix H.

We start with a trivial example where two agents are independent, i.e. each following independent MDPs. It's clear that the global value function can be decomposed as the sum of value functions of local MDPs. As two agents are independent, it holds $P^\pi(s'_A, s'_B \mid s_A, s_B) = P^\pi(s'_A \mid s_A) \cdot P^\pi(s'_B \mid s_B)$, or in matrix form,

$$\boldsymbol{P}^\pi_{AB} = \boldsymbol{P}^\pi_A \otimes \boldsymbol{P}^\pi_B \,,$$

where $\otimes$ is the tensor product or Kronecker product of matrices. The important question is whether we can extend beyond this trivial case of independent subsystems.

**A Sufficient and Necessary Condition** We introduce a new condition called "Markov Entanglement" to describe the intrinsic structure of transition dynamics in multi-agent MDPs.

> **Definition 1** (Markov Entanglement). *Consider a two-agent MDP with transition $\boldsymbol{P}^\pi_{AB}$. If there exists*
> $$\boldsymbol{P}^\pi_{AB} = \sum_{j=1}^{K} x_j \boldsymbol{P}^{(j)}_A \otimes \boldsymbol{P}^{(j)}_B \,,$$
> *then $\boldsymbol{P}^\pi_{AB}$ is separable; otherwise entangled.*

Compared with the preceding example of independent subsystems, Markov entanglement offers an intuitive interpretation: a two-agent MDP is separable if it can be expressed as a *linear combination of independent subsystems*. We then demonstrate,

$$\text{separable } \boldsymbol{P}^\pi_{AB} \iff \text{ decomposable } \boldsymbol{V}^\pi_{AB} \,,$$

where $\boldsymbol{V}^\pi_{AB}$ is decomposable if there exist local value functions $\boldsymbol{V}_A, \boldsymbol{V}_B$ such that $V^\pi_{AB}(s_A, s_B) = V_A(s_A) + V_B(s_B)$ for all $(s_A, s_B)$. This result sharply unravels the secret structure of system dynamics governing value decomposition. As a sufficient condition, our finding strictly generalizes the previous independent subsystem example, extending it to scenarios involving interacting and coupled agents. As a necessary condition, we prove that exact value decomposition under any reward kernel requires the system dynamics to be separable. Taken together, this result provides a *complete characterization* of when exact value function decomposition is possible in multi-agent MDPs.

More interestingly, our Markov entanglement condition turns out be a mathematical counterpart of quantum entanglement in quantum physics, whose definition is provided below.

> **Definition 2** (Quantum Entanglement). *Consider a two-party quantum state $\rho_{AB}$. If there exists*
>
> $$\rho_{AB} = \sum_{j=1}^{K} x_j \rho_A^{(j)} \otimes \rho_B^{(j)}, \quad \boldsymbol{x} \geq 0,$$
>
> *then $\rho_{AB}$ is separable; otherwise entangled.*

The quantum state is represented by a *density matrix*, a positive semidefinite matrix with unit trace, analogous to transition matrix in the Markov world. The concept of quantum entanglement describes the inter-dependence of particles in a quantum system, while Markov entanglement describes that of agents in a Markov system.

Finally, we introduce several novel proof techniques concerning the sufficient and necessary condition, including an "absorbing" technique for separable transition matrices and a novel characterization of the linear space spanned by tensor products of transition matrices. We believe these techniques hold independent interest for the broader RL community.

**Decomposition Error in General Multi-agent MDPs**  Despite the precise characterization of Markov entanglement and exact value decomposition, general multi-agent MDPs can exhibit arbitrary complexity, with agents intricately entangled. This raises a critical question: *can value decomposition serve as a meaningful approximation in such scenarios?* To address this, we introduce a mathematical quantification to measure the Markov entanglement in general multi-agent MDPs,

$$E(\boldsymbol{P}_{AB}^{\pi}) \coloneqq \min_{\boldsymbol{P} \in \mathcal{P}_{\text{SEP}}} d(\boldsymbol{P}_{AB}^{\pi}, \boldsymbol{P}), \tag{1}$$

where $\mathcal{P}_{\text{SEP}}$ is the set of all separable transition matrices and $d(\cdot, \cdot)$ is some distance measure. In other words, the degree of Markov entanglement is determined by its distance to the closest separable transition matrix. This concept can also find its counterpart in quantum physics, with the measure of quantum entanglement defined as

$$E(\rho_{AB}) \coloneqq \min_{\rho \in \rho_{\text{SEP}}} d(\rho_{AB}, \rho),$$

where $\rho_{\text{SEP}}$ is the set of all separable quantum states. In quantum physics, various distance measures have been designed for density matrices and capture different physical interpretations [31]. In the Markov world, we analogously design distance measures for transition matrices and relate them to the value decomposition error,

$$\left\| \text{decomposition error of } \boldsymbol{V}_{AB}^{\pi} \right\| = \mathcal{O}\left( E(\boldsymbol{P}_{AB}^{\pi}) \right).$$

where $\|\cdot\|$ depends on the distance we use to measure Markov entanglement. We explore diverse distance measures including the well-known total variation distance and its stationary distribution weighted variant. We also design a novel agent-wise distance incorporating the multi-agent structure, which may be of independent interest to the MARL community. We further demonstrate how different distance measures give birth to the decomposition error in different norms.

**Applications of Markov Entanglement**  Finally, we leverage our Markov entanglement theory to analyze several structured multi-agent MDPs. We prove that a widely-used class of index policies is asymptotically separable, exhibiting a decomposition error that scales as $\mathcal{O}(\sqrt{N})$ with the number of agents $N$. This result theoretically justifies the practical effectiveness of value decomposition for index-based policies. Our proof builds on innovations that integrate Markov entanglement with mean-field analysis. We also show that Markov entanglement admits an efficient empirical estimation, thus helping practitioners determine when value decomposition is feasible.

## 1.2 Other related work

In the first section, we have reviewed typical empirical works on value decomposition. Here, we complement that discussion with related literature on theoretical insights.

Prior theoretical research has extensively investigated the decomposition of optimal value functions in multi-agent settings. A prominent area involves Lagrange relaxation, with the Restless Multi-Armed

Bandit (RMAB, [49]) as a foundational model. Lagrange relaxation decouples the constraint of agents, yielding a decomposable value that upper bounds the original value. The per-agent decomposition error is proven to decay asymptotically to zero [47, 48, 41] and enjoys a quadratic or exponential rate [20, 21, 11, 51, 52]. Other work generalizes to Weakly-Coupled MDPs (WCMDPs) [6, 13, 19]. However, [2] showed Lagrange relaxation can have arbitrarily large errors and proposed an alternative decomposition called Approximate Linear Programs (ALP), which is proven to have tighter error [12]. Despite these advancements, characterizing decomposition error for general multi-agent MDPs remains unknown. In contrast, our Markov entanglement theory analyzes value decomposition for general multi-agent MDPs under arbitrary policies, including optimal ones.

Another line of theoretical work has concentrated on policy optimization via value decomposition. Despite reported empirical successes, rigorous theoretical analysis remains challenging. [5] derived an approximation ratio for a specific index policy on a two-state RMAB. [43, 16] analyzed the convergence of the CTDE paradigm under strong exploration assumptions, while also highlighting scenarios of divergence. In contrast, our work instead focuses on policy evaluation rather than optimization. This enables us to derive clear and interpretable bounds on the decomposition error for general finite-state multi-agent MDPs that only require the existence of a stationary distribution.

**Notations**   We abbreviate subscripts $(\boldsymbol{s}) \coloneqq (s_{1:N}) \coloneqq (s_1, s_2, \ldots, s_N)$. Particularly, for two-agent case, when the context is clear, we abbreviate $(\boldsymbol{s}) \coloneqq (s_{AB}) \coloneqq (s_A, s_B)$. Let $[N] = \{1, 2, \ldots, N\}$ and $\mathbb{Z}^+$ be the set of positive integers.

## 2   Model

We consider a standard two-agent MDP $\mathcal{M}_{AB}(\mathcal{S}, \mathcal{A}, \boldsymbol{P}, \boldsymbol{r}_A, \boldsymbol{r}_B, \gamma)$ with joint state space $\mathcal{S} = \mathcal{S}_A \times \mathcal{S}_B$ and joint action space $\mathcal{A} = \mathcal{A}_A \times \mathcal{A}_B$ where $A, B$ represent two agents. For simplicity, let $|\mathcal{S}_A| = |\mathcal{S}_B| = |S|$ and $|\mathcal{A}_A| = |\mathcal{A}_B| = |A|$. For agents at global state $\boldsymbol{s} = (s_A, s_B)$ with action $\boldsymbol{a} = (a_A, a_B)$ taken, the system will transit to $\boldsymbol{s}' = (s_A', s_B')$ according to transition kernel $\boldsymbol{s}' \sim \boldsymbol{P}(\cdot \mid \boldsymbol{s}, \boldsymbol{a})$ and each agent $i \in \{A, B\}$ will receive its local reward $r_i(s_i, a_i)$. The global reward $r_{AB}$ is defined as the summation of local rewards $r_{AB}(\boldsymbol{s}, \boldsymbol{a}) \coloneqq r_A(s_A, a_A) + r_B(s_B, a_B)$, or in vector form $\boldsymbol{r}_{AB} \in \mathbb{R}^{|S|^2|A|^2} \coloneqq \boldsymbol{r}_A \otimes \boldsymbol{e} + \boldsymbol{e} \otimes \boldsymbol{r}_B$, where $\otimes$ is the tensor product and $\boldsymbol{e} = \boldsymbol{1} \in \mathbb{R}^{|S||A|}$ is the vector of all ones.[1] We further assume the local rewards are bounded, i.e. for agent $i \in \{A, B\}$, $|r_i(s_i, a_i)| \leq r_{\max}^i$ for all $(s_i, a_i)$.

Given any global policy $\pi \colon \mathcal{S} \to \Delta(\mathcal{A})$, the global Q-value under policy $\pi$ is defined as the discounted summation of global rewards $Q_{AB}^\pi(\boldsymbol{s}, \boldsymbol{a}) = \mathbb{E}\left[\sum_{t=0}^\infty \gamma^t r_{AB}(\boldsymbol{s}^t, \boldsymbol{a}^t) \mid \pi, (\boldsymbol{s}^0, \boldsymbol{a}^0) = (\boldsymbol{s}, \boldsymbol{a})\right]$ where $\gamma \in [0, 1)$ is the discount factor. The value function is then defined as $V_{AB}^\pi(\boldsymbol{s}) = \mathbb{E}_{\boldsymbol{a} \sim \pi(\cdot \mid \boldsymbol{s})}[Q_{AB}^\pi(\boldsymbol{s}, \boldsymbol{a})]$. We denote $\boldsymbol{P}_{AB}^\pi \in \mathbb{R}^{|S|^2|A|^2 \times |S|^2|A|^2}$ as the transition matrix induced by $\pi$ where $P_{AB}^\pi(\boldsymbol{s}', \boldsymbol{a}' \mid \boldsymbol{s}, \boldsymbol{a}) = \boldsymbol{P}(\boldsymbol{s}' \mid \boldsymbol{s}, \boldsymbol{a}) \cdot \pi(\boldsymbol{a}' \mid \boldsymbol{s}')$. Then by the Bellman Equation, we have $Q_{AB}^\pi = (\boldsymbol{I} - \gamma \boldsymbol{P}_{AB}^\pi)^{-1} \boldsymbol{r}_{AB}$. Our objective is to decompose this global Q-value $Q_{AB}^\pi$ as the summation of some local functions $Q_A$ and $Q_B$, i.e. $Q_{AB}^\pi(\boldsymbol{s}, \boldsymbol{a}) = Q_A(s_A, a_A) + Q_B(s_B, a_B)$, or in vector form,

$$Q_{AB}^\pi = Q_A \otimes \boldsymbol{e} + \boldsymbol{e} \otimes Q_B. \tag{2}$$

Notice we formally introduce our research question using Q-value instead of V-value function as in the introduction. Q-value decomposition is a stronger result that implies V-value function decomposition. It also turns out that Q-value further incorporates action information enabling more general theoretical analysis. More discussions can be found in Appendix B.

### 2.1   Local (Q-)value functions

Recent literature offers several algorithms for learning local (Q-)values. In this paper, we use a meta-algorithm framework in 1 to summarize their underlying principles.

This meta-algorithm framework is simple and intuitive: each agent independently fits its local Q-values based on its local observations. Notably, the framework requires no prior knowledge of the MDP, and learning can be performed in a fully decentralized manner. Furthermore, we use term *meta* in that we do not pose restrictions on how agents estimate their local Q-values. For tabular case, one

---

[1]In Appendix J.4, we extend our results to multi-agent MDP model where the global cannot be decomposed.

---

**Meta Algorithm 1:** Leaning Local Q-value Functions

---

**Require:** Global policy $\pi$; horizon length $T$.

1: Execute $\pi$ for $T$ epochs and obtain $\mathcal{D} = \left\{ (s_{AB}^t, a_{AB}^t, r_{AB}^t, s_{AB}^{t+1}, a_{AB}^{t+1}) \right\}_{t=1}^{T-1}$.

2: Each agent $i \in \{A, B\}$ fits $Q_i^\pi$ using local observations $\mathcal{D}_i = \left\{ (s_i^t, a_i^t, r_i^t, s_i^{t+1}, a_i^{t+1}) \right\}_{t=1}^{T-1}$.

---

can plug in Temporal Difference (TD) learning [39] or its variants. For large-scale problems, one can apply linear function approximations (e.g. [5, 24, 8]) or more sophisticated neural networks (e.g. [33, 38, 29]).

Despite the flexibility in fitting local value functions, it is helpful to call out a particular approach: TD learning for local Q-values in the tabular case, as it facilitates the analysis and reveals the structure of value decomposition in the next section.

**Local TD learning.** Although each agent's environment is not Markovian in a local sense (it is, more precisely, partially observed Markovian), one can still define its "marginalized" local transition matrix under the stationary distribution. Mathematically, for agent $A$, we denote $\boldsymbol{P}_A^\pi \in \mathbb{R}^{|S||A| \times |S||A|}$ as its local transition where

$$P_A^\pi(s_A', a_A' \mid s_A, a_A) = \sum_{s_B', a_B'} \sum_{s_B, a_B} P_{AB}^\pi \left( s_{AB}', a_{AB}' \mid s_{AB}, a_{AB} \right) \mu_{AB}^\pi(s_B, a_B \mid s_A, a_A). \quad (3)$$

Here, $\mu_{AB}^\pi \in \Delta(\mathcal{S})$ denotes the global stationary distribution under policy $\pi$ (for convenience, we assume $\pi$ induces a unichain, i.e. $\mu_{AB}^\pi$ is unique and strictly positive).[2] Given this "marginalized" local transition, the local Q-values obtained by Meta Algorithm 1 using tabular TD learning converge to the solution of the following "marginalized" Bellman equation:

$$Q_A^\pi = (\boldsymbol{I} - \gamma \boldsymbol{P}_A^\pi)^{-1} \, \boldsymbol{r}_A \,.$$

By symmetry, we can derive analogous results for agent $B$, obtaining its transition matrix $\boldsymbol{P}_B^\pi$ and local Q-values $Q_B^\pi$. Next, we show how $Q_A^\pi$ and $Q_B^\pi$ contribute to the exact value decomposition.

## 3   Exact value decomposition

To begin, recall the key condition we identify in the introduction: *Markov Entanglement* in Definition 1. Our first theorem shows that an MDP with no Markov entanglement is indeed sufficient for the exact value decomposition. More importantly, local TD learning (or Meta Algorithm 1 more generally) is guaranteed to recover such decomposition, i.e. $Q_{AB}^\pi = Q_A^\pi \otimes \boldsymbol{e} + \boldsymbol{e} \otimes Q_B^\pi$.

**Theorem 1.** *Consider a two-agent MDP $\mathcal{M}_{AB}$ and policy $\pi \colon \mathcal{S} \to \Delta(\mathcal{A})$. If two agents are separable, i.e. there exists $K \in \mathbb{Z}^+$, measure $\{x_j\}_{j \in [K]}$, and transition matrices $\left\{ \boldsymbol{P}_A^{(j)}, \boldsymbol{P}_B^{(j)} \right\}_{j \in [K]}$ such that $\boldsymbol{P}_{AB}^\pi = \sum_{j=1}^K x_j \boldsymbol{P}_A^{(j)} \otimes \boldsymbol{P}_B^{(j)}$. Then it holds $\boldsymbol{P}_A^\pi = \sum_{i=1}^K x_j \boldsymbol{P}_A^{(j)}$ and $\boldsymbol{P}_B^\pi = \sum_{j=1}^K x_j \boldsymbol{P}_B^{(j)}$. Furthermore, the Eq. (2) holds*

$$Q_{AB}^\pi = Q_A^\pi \otimes \boldsymbol{e} + \boldsymbol{e} \otimes Q_B^\pi \,.$$

This theorem establishes that even when the system is not independent, as long as it can be represented as a *linear combination of independent subsystems*, the global Q-value admits an exact decomposition.

**An illustrative example of coupling and Markov entanglement**   To elucidate the concept of Markov entanglement, we present an example of two-agent MDP where agents are coupled but not entangled. Consider a two-agent MDP $\mathcal{M}_{AB}$ with $|\mathcal{A}_A| = |\mathcal{A}_B| = 2$, where action 1 means activate and 0 means idle. Each agent $i \in \{A, B\}$ has its own local transition kernel $\boldsymbol{P}_i$. We examine the following policy: at each time-step, we randomly activate one agent and keep another idle, i.e.

---

[2]For $\mu_{AB}^\pi(s_B, a_B \mid s_A, a_A)$ to be well-defined, we require $\mu_{AB}^\pi(s_A, a_A) > 0$. If $\mu_{AB}^\pi(s_A, a_A) = 0$, then action $a_A$ is never taken in state $s_A$ under policy $\pi$, and we exclude such pairs by restricting the feasible action set $\mathcal{A}(s_A)$. All theoretical results apply to the remaining valid state-action pairs.

$\pi(\boldsymbol{a} \mid \boldsymbol{s}) = 1/2$ if $\boldsymbol{a} = (0,1)$ or $\boldsymbol{a} = (1,0)$. Consequently, this policy couples the agents through the constraint $a_A + a_B = 1$ at each timestep. However, we will demonstrate that despite this coupling, there's *no* entanglement. Specifically, we construct the following decomposition

$$\boldsymbol{P}_{AB}^{\pi} = \frac{1}{2}\boldsymbol{P}_A^0 \otimes \boldsymbol{P}_B^1 + \frac{1}{2}\boldsymbol{P}_A^1 \otimes \boldsymbol{P}_B^0 \,, \tag{4}$$

where $\boldsymbol{P}_i^a$ refers to the transition matrix of agents $i \in \{A, B\}$ taking action $a \in \{0, 1\}$. Intuitively, the right-hand side of Eq. (4) describes how at each time step, the global system randomly selects between two possible transitions: $\boldsymbol{P}_A^0 \otimes \boldsymbol{P}_B^1$ or $\boldsymbol{P}_A^1 \otimes \boldsymbol{P}_B^0$, each with equal probability (akin to rolling a fair dice). This example thus clearly demonstrates a *coupled* system can still be *separable* and thus admits an exact value decomposition.

**Proof of sufficiency** Theorem 1 admits a simple proof based on the several basic properties of tensor product. First of all, given $\boldsymbol{P}_{AB}^{\pi} = \sum_{j=1}^{K} x_j \boldsymbol{P}_A^{(j)} \otimes \boldsymbol{P}_B^{(j)}$, we can plug this into the formulation of $\boldsymbol{P}_A^{\pi}$ in Eq. (3) and quickly verify $\boldsymbol{P}_A^{\pi} = \sum_{i=1}^{K} x_i \boldsymbol{P}_A^{(i)}$. It remains to show Eq. (2). Notice that

$$(\boldsymbol{I} - \gamma \boldsymbol{P}_{AB}^{\pi})^{-1} (\boldsymbol{r}_A \otimes \boldsymbol{e}) = \sum_{t=0}^{\infty} \gamma^t \left( \sum_{j=1}^{K} x_j \boldsymbol{P}_A^{(j)} \otimes \boldsymbol{P}_B^{(j)} \right)^t (\boldsymbol{r}_A \otimes \boldsymbol{e})$$

$$\overset{(i)}{=} \sum_{t=0}^{\infty} \gamma^t \left( \left( \sum_{j=1}^{K} x_j \boldsymbol{P}_A^{(j)} \right)^t \boldsymbol{r}_A \right) \otimes \boldsymbol{e} = \left( (\boldsymbol{I} - \gamma \boldsymbol{P}_A^{\pi})^{-1} \boldsymbol{r}_A \right) \otimes \boldsymbol{e} = Q_A^{\pi} \otimes \boldsymbol{e} \,.$$

where we refer to $(i)$ as an "absorbing" technique based on the bilinearity and mixed-product property of tensor product[3]. Specifically, since $\boldsymbol{P}\boldsymbol{e} = \boldsymbol{e}$ for any transition matrix $\boldsymbol{P}$, we have for any $t$,

$$\left( \sum_{j=1}^{K} x_j \boldsymbol{P}_A^{(j)} \otimes \boldsymbol{P}_B^{(j)} \right)^t (\boldsymbol{r}_A \otimes \boldsymbol{e}) = \left( \sum_{j=1}^{K} x_j \boldsymbol{P}_A^{(j)} \otimes \boldsymbol{P}_B^{(j)} \right)^{t-1} \left( \sum_{j=1}^{K} x_j \left( \boldsymbol{P}_A^{(j)} \boldsymbol{r}_A \right) \otimes \left( \boldsymbol{P}_B^{(j)} \boldsymbol{e} \right) \right)$$

$$= \left( \sum_{j=1}^{K} x_j \boldsymbol{P}_A^{(j)} \otimes \boldsymbol{P}_B^{(j)} \right)^{t-1} \left( \sum_{j=1}^{K} x_j \boldsymbol{P}_A^{(j)} \boldsymbol{r}_A \right) \otimes \boldsymbol{e} = \ldots = \left( \left( \sum_{j=1}^{K} x_j \boldsymbol{P}_A^{(j)} \right)^t \boldsymbol{r}_A \right) \otimes \boldsymbol{e} \,.$$

Similar results can be derived for $\boldsymbol{P}_B^{\pi}$ such that $(\boldsymbol{I} - \gamma \boldsymbol{P}_{AB}^{\pi})^{-1} (\boldsymbol{e} \otimes \boldsymbol{r}_B) = \boldsymbol{e} \otimes Q_B^{\pi}$. Finally, combining the above results, we have

$$Q_{AB}^{\pi} = (\boldsymbol{I} - \gamma \boldsymbol{P}_{AB}^{\pi})^{-1} \boldsymbol{r}_{AB} = (\boldsymbol{I} - \gamma \boldsymbol{P}_{AB}^{\pi})^{-1} (\boldsymbol{r}_A \otimes \boldsymbol{e} + \boldsymbol{e} \otimes \boldsymbol{r}_B) = Q_A^{\pi} \otimes \boldsymbol{e} + \boldsymbol{e} \otimes Q_B^{\pi} \,.$$

### 3.1 Necessary condition for the exact value decomposition

We then investigate whether Markov entanglement is necessary for the exact Q-value decomposition. The answer is in general no, since one can construct trivial counterexamples such as $\boldsymbol{r}_A = \boldsymbol{r}_B = \boldsymbol{0}$ or $\gamma = 0$, where the decomposition trivially holds. On the other hand, we focus on a stronger and more general concept of the exact value decomposition that holds under any reward kernel given $\gamma > 0$. Formally, we present the following theorem.

**Theorem 2.** *Consider a two-agent Markov MDP $\mathcal{M}_{AB}$ with discount factor $\gamma > 0$ and $\pi \colon \mathcal{S} \to \Delta(\mathcal{A})$. Suppose there exists local functions $Q_i \colon \boldsymbol{r}_i \to \mathbb{R}^{|S||A|}$ for $i \in \{A, B\}$ such that $Q_{AB}^{\pi} = Q_A(\boldsymbol{r}_A) \otimes \boldsymbol{e} + \boldsymbol{e} \otimes Q_B(\boldsymbol{r}_B)$ holds for any pair of reward $\boldsymbol{r}_A, \boldsymbol{r}_B$, then $A, B$ must be separable.*

Combined with Theorem 1, we conclude Markov entanglement serves as a sufficient and necessary condition for the exact value decomposition. We also emphasize that Theorem 2 considers general local functions $Q_i$. This generality accommodates all methods for fitting local $Q_i$, such as deep neural networks, provided that the training relies solely on the local observations of agent $i$.

There exist other possible ways for value decomposition. For example, [38, 16] consider $Q_{AB}^{\pi}(\boldsymbol{s}, \boldsymbol{a}) = L_A(s_A, a_A, \boldsymbol{r}_{AB}) + L_B(s_B, a_B, \boldsymbol{r}_{AB})$ where $L_A, L_B$ are learned jointly via minimizing the global Bellman error[4]; [35, 29, 37, 42] consider general monotonic operations beyond

---

[3]We introduce several basic properties of tensor product in Appendix A.

[4]In Appendix E, we provide an example of entangled MDP that allows for an exact value decomposition where $L_A$ depends on both $\boldsymbol{r}_A$ and $\boldsymbol{r}_B$.

additive decompositions. These methods introduce possibly richer representations at the cost of more sophisticated implementations and less interpretability, which is beyond the scope of this paper.

**Proof sketch of necessity** Our proof builds on several novel techniques. Recall $\mathcal{P}_{\text{SEP}}$ is the set of all separable transition matrices.

**Step 1: Understanding the orthogonal complement.** If a transition matrix is entangled, it will have non-zero component in the orthogonal complement of $\mathcal{P}_{\text{SEP}}$, which we construct as

$$\mathcal{P}_{\text{SEP}}^{\perp} = \left\{ \sum_{j=1}^{|S||A|-1} \left( \varepsilon_j \boldsymbol{e}^{\top} \right) \otimes \boldsymbol{W}_j^1 + \sum_{j=1}^{|S||A|-1} \boldsymbol{W}_j^2 \otimes \left( \varepsilon_j \boldsymbol{e}^{\top} \right) \middle| W_{1:j}^1, W_{1:j}^2 \in \mathbb{R}^{|S||A| \times |S||A|} \right\},$$

where $\varepsilon_j = (1, 0, \dots, 0, -1, 0, \dots, 0)^{\top}$ with the first element 1 and $(j+1)$-th element $-1$. Then, we study an intermediate transition matrix $(1 - \gamma)(\boldsymbol{I} - \gamma \boldsymbol{P}_{AB}^{\pi})^{-1}$. We show if it's entangled, we are able to construct $\boldsymbol{r}_A, \boldsymbol{r}_B$ based on its component in $\mathcal{P}_{\text{SEP}}^{\perp}$ such that $Q_{AB}^{\pi}$ is not decomposable under this pair of rewards. We thus conclude decomposable $Q_{AB}^{\pi} \implies$ separable $(1 - \gamma)(\boldsymbol{I} - \gamma \boldsymbol{P}_{AB}^{\pi})^{-1}$.

**Step 2: Connecting to "inverse".** Finally, we complete the proof via a lemma showing separable $(1 - \gamma)(\boldsymbol{I} - \gamma \boldsymbol{P}_{AB}^{\pi})^{-1} \iff$ separable $\boldsymbol{P}_{AB}^{\pi}$. The $\impliedby$ side is straightforward since $(\boldsymbol{I} - \gamma \boldsymbol{P}_{AB}^{\pi})^{-1}$ is the Neumann series of $\gamma \boldsymbol{P}_{AB}^{\pi}$. For the converse $\implies$, we seek to invert this Neumann series. This is achieved by a careful analysis of the operator norm of $\boldsymbol{I} - (1 - \gamma)(\boldsymbol{I} - \gamma \boldsymbol{P}_{AB}^{\pi})^{-1}$.

## 4 Value decomposition error in general two-agent MDPs

In general, the system transition $\boldsymbol{P}_{AB}^{\pi}$ can be arbitrarily entangled. In these scenarios, we investigate when value decomposition $Q_A^{\pi} \otimes \boldsymbol{e} + \boldsymbol{e} \otimes Q_B^{\pi}$ is an effective approximation of $Q_{AB}^{\pi}$. As mentioned in the introduction, we define the measure of Markov entanglement in Eq. (1) as certain distance between $\boldsymbol{P}_{AB}^{\pi}$ and its closet separable transition matrix. We will examine several distance measures for transition matrices and relate them to the decomposition error.

### 4.1 Entry-wise error bound

**Total variation distance** One widely used metric for transition matrices is Total Variation (TV) distance. Specifically, for two transition matrices $\boldsymbol{P}, \boldsymbol{P}' \in \mathbb{R}^{|S|^2|A|^2 \times |S|^2|A|^2}$, define

$$\|\boldsymbol{P} - \boldsymbol{P}'\|_{\text{TV}} := \max_{(\boldsymbol{s}, \boldsymbol{a}) \in \mathcal{S} \times \mathcal{A}} D_{\text{TV}}\left( \boldsymbol{P}(\cdot \mid \boldsymbol{s}, \boldsymbol{a}), \boldsymbol{P}'(\cdot \mid \boldsymbol{s}, \boldsymbol{a}) \right), \tag{5}$$

where $D_{\text{TV}}$ is the total variation distance between probability measures. While TV distance is straightforward, it does not take into account the inherent multi-agent structure.

**Agent-wise distance** We thus introduce a more refined distance specially designed for multi-agent MDPs. Formally, the Agent-wise Total Variation (ATV) distance between two transition matrices $\boldsymbol{P}, \boldsymbol{P}' \in \mathbb{R}^{|S|^2|A|^2 \times |S|^2|A|^2}$ w.r.t agent $A$ is defined as

$$\|\boldsymbol{P} - \boldsymbol{P}'\|_{\text{ATV}_A} := \max_{(\boldsymbol{s}, \boldsymbol{a}) \in \mathcal{S} \times \mathcal{A}} D_{\text{TV}}\left( \sum_{s_B', a_B'} \boldsymbol{P}(\cdot, \cdot \mid \boldsymbol{s}, \boldsymbol{a}), \sum_{s_B', a_B'} \boldsymbol{P}'(\cdot, \cdot \mid \boldsymbol{s}, \boldsymbol{a}) \right). \tag{6}$$

The ATV distance w.r.t agent $B$ can be defined similarly. Intuitively, compared to TV, ATV focuses on an individual agent and measures the difference between its local transitions. One can also verify ATV is tighter distance, i.e. $\|\boldsymbol{P} - \boldsymbol{P}'\|_{\text{ATV}_A} \leq \|\boldsymbol{P} - \boldsymbol{P}'\|_{\text{TV}}$. We can plug ATV into Eq. (1) and obtain the measure of Markov entanglement w.r.t ATV distance $E_i(\boldsymbol{P}_{AB}^{\pi}) := \min_{\boldsymbol{P} \in \mathcal{P}_{\text{SEP}}} \|\boldsymbol{P}_{AB}^{\pi} - \boldsymbol{P}\|_{\text{ATV}_i}$ for $i \in \{A, B\}$. In fact, one can also verify

$$E_A(\boldsymbol{P}_{AB}^{\pi}) = \min_{\boldsymbol{P}_A} \max_{(\boldsymbol{s}, \boldsymbol{a}) \in \mathcal{S} \times \mathcal{A}} D_{\text{TV}}\left( \boldsymbol{P}_{AB}^{\pi}(\cdot, \cdot \mid \boldsymbol{s}, \boldsymbol{a}), \boldsymbol{P}_A(\cdot, \cdot \mid s_A, a_A) \right), \tag{7}$$

The following theorem connects these measures to the value decomposition error.

**Theorem 3.** *Consider a two-agent Markov system $\mathcal{M}_{AB}$ and policy $\pi \colon \mathcal{S} \to \Delta(\mathcal{A})$ with the measure of Markov entanglement $E_A(\boldsymbol{P}_{AB}^\pi), E_B(\boldsymbol{P}_{AB}^\pi)$ defined in Eq. (7), then the decomposition error is entry-wise bounded by the measure of Markov entanglement,*

$$\left\| Q_{AB}^\pi - (Q_A^\pi \otimes \boldsymbol{e} + \boldsymbol{e} \otimes Q_B^\pi) \right\|_\infty \leq \frac{4\gamma \left( E_A(\boldsymbol{P}_{AB}^\pi) r_{\max}^A + E_B(\boldsymbol{P}_{AB}^\pi) r_{\max}^B \right)}{(1-\gamma)^2} .$$

## 4.2 Error weighted by stationary distribution

Entry-wise error bound is a very strong result for Q-value decomposition. This comes with the entry-wise TV bounds in both TV and ATV distance. An alterative choice is to consider an error weighted by the stationary distribution. Formally, consider

$$\left\| Q_{AB}^\pi - (Q_A^\pi \otimes \boldsymbol{e} + \boldsymbol{e} \otimes Q_B^\pi) \right\|_{\mu_{AB}^\pi} := \sum_{\boldsymbol{s},\boldsymbol{a}} \mu_{AB}^\pi(\boldsymbol{s},\boldsymbol{a}) \Big| Q_{AB}^\pi(\boldsymbol{s},\boldsymbol{a}) - (Q_A^\pi(s_A, a_A) + Q_B^\pi(s_B, a_B)) \Big| .$$

We note that this norm is clearly weaker than the entry-wise norm. Nevertheless, a stationary distribution weighted error bound is sufficient in many practical scenarios. Similar ideas are also quite common in policy evaluation literature [14, 40, 9].

**Distance weighted by stationary distribution**   To analyze this $\mu_{AB}^\pi$-weight decomposition error, we analogously propose the $\mu_{AB}^\pi$-weighted distance measure of Markov entanglement. Specifically, we have the following $\mu_{AB}^\pi$-weighted version of Eq. (7).

$$E_A(\boldsymbol{P}_{AB}^\pi) = \min_{\boldsymbol{P}_A} \sum_{\boldsymbol{s},\boldsymbol{a}} \mu_{AB}^\pi(\boldsymbol{s},\boldsymbol{a}) D_{\mathrm{TV}} \Big( \boldsymbol{P}_{AB}^\pi(\cdot,\cdot \mid \boldsymbol{s},\boldsymbol{a}), \boldsymbol{P}_A(\cdot,\cdot \mid s_A, a_A) \Big) . \tag{8}$$

Eq. (8) substitutes the $\mu_{AB}^\pi$-weighted average for the maximum operator in Eq. (7). Finally, we have the following variant of Theorem 3.

**Theorem 4.** *Under the same setup as Theorem 3 with $\mu_{AB}^\pi$-weighted measure of Markov entanglement $E_A(\boldsymbol{P}_{AB}^\pi), E_B(\boldsymbol{P}_{AB}^\pi)$ defined in Eq. (8), the $\mu_{AB}^\pi$-weighted decomposition error is bounded,*

$$\left\| Q_{AB}^\pi - (Q_A^\pi \otimes \boldsymbol{e} + \boldsymbol{e} \otimes Q_B^\pi) \right\|_{\mu_{AB}^\pi} \leq \frac{4\gamma \left( E_A(\boldsymbol{P}_{AB}^\pi) r_{\max}^A + E_B(\boldsymbol{P}_{AB}^\pi) r_{\max}^B \right)}{(1-\gamma)^2} .$$

Compared to Theorem 3, Theorem 4 measures a weaker $\mu_{AB}^\pi$-weighted decomposition error, while the condition on $\boldsymbol{P}_{AB}^\pi$ is also relaxed, requiring only a weighted average bound in Eq. (8).

## 4.3 Multi-agent Markov entanglement

Finally, we extend the results to multi-agent MDPs with the measure of Markov entanglement $E_{1:N}(\boldsymbol{P}_{1:N}^\pi)$ for an $N$-agent MDP. The extension is relatively straightforward. We demonstrate the extension of Theorem 4 below and more details can be found in Appendix H.

**Theorem 5.** *Consider a $N$-agent MDP $\mathcal{M}_{1:N}$ with the measure of Markov entanglement $E_i(\boldsymbol{P}_{1:N}^\pi)$ w.r.t ATV distance, the $\mu_{1:N}^\pi$-weighted decomposition error is bounded by the measure of Markov entanglement,*

$$\left\| Q_{1:N}^\pi(\boldsymbol{s},\boldsymbol{a}) - \sum_{i=1}^N Q_i^\pi(s_i, a_i) \right\|_{\mu_{1:N}^\pi} \leq \frac{4\gamma \left( \sum_{i=1}^N E_i(\boldsymbol{P}_{1:N}^\pi) r_{\max}^i \right)}{(1-\gamma)^2} .$$

# 5  Applications of Markov Entanglement

In this section, we apply Markov entanglement and demonstrate a widely-used class of index policies is asymptotically separable. To begin, we introduce the model of Restless Multi-Armed Bandit (RMAB, [49]). In an $N$-agent RMAB, each agent follows a homogeneous two-action MDP with action 1 meaning activate and 0 idle. A central decision maker will activate $M \leq N$ agents at each timestep and leave other agents idle. In other words, agents transit independently but are coupled under constraint $\sum_{i=1}^N a_i = M$. In RMAB, arguably the most classical and widely-used policy is the index policy, which we formally define as

**Definition 3** (Index Policy). *There exists a priority index $\nu_s$ for each local state $s$. The decision maker will always activate agents in the descending order of the priority until the budget constraint $M$ is met. Ties are resolved fairly via uniform random sampling of agents at the same state.*

The index policy traces back to the well-known Gittins Index [46], Whittle Index [49, 47, 20], and fluid-based index policies [41, 21]. [33, 4, 5, 30, 44, 3] apply data-driven method to optimize index policies and report great empirical success in industrial implementations. Understanding the mystery behind such success calls for a theory for general index policies. We then present our main theorem.

**Theorem 6.** *Consider an $N$-agent restless multi-armed bandit. For any index policy satisfying mild technical conditions, there exists constant $C$ independent of $N$, such that for any agent $i \in [N]$, its $\mu_{1:N}^\pi$-weighted measure of Markov entanglement is bounded, $E_i(\boldsymbol{P}_{1:N}^\pi) \le C/\sqrt{N}$.*

Theorem 6 requires two standard technical conditions for index policies: non-degenerate and uniform global attractor property, which are used in almost all related theoretical work [47, 41, 20, 21] and are detailed in Appendix I. Theorem 6 justifies index polices are asymptotically separable. Combined with an $N$-agent version of Theorem 4, we obtain the sublinear decomposition error for index policies

$$\left\| Q_{1:N}^\pi(\boldsymbol{s}, \boldsymbol{a}) - \sum_{i=1}^{N} Q_i^\pi(s_i, a_i) \right\|_{\mu_{1:N}^\pi} \le \mathcal{O}(\sqrt{N}).$$

This sublinear error result explains why the value decomposition in [33, 4, 5] manages to effectively approximate the global value function in large-scale practical applications.

### 5.1 Efficient verification of value decomposition

For practitioners, verifying the feasibility of value decomposition is challenging due to the exponential computational complexity of estimating the global Q-value. As a solution, Markov entanglement offers an efficient way to empirically test whether value decomposition can be safely applied. Consider the $\mu_{AB}^\pi$-weighted measure of Markov entanglement in Eq. (8), we have

$$E_A(\boldsymbol{P}_{AB}^\pi) \approx \frac{1}{2} \min_{\boldsymbol{P}_A} \frac{1}{T} \sum_{t=1}^{T} \sum_{s_A', a_A'} \left| \boldsymbol{P}_{AB}^\pi(s_A', a_A' \mid \boldsymbol{s}^t, \boldsymbol{a}^t) - \boldsymbol{P}_A(s_A', a_A' \mid s_A^t, a_A^t) \right| \tag{9}$$

In other words, we can apply a Monte-Carlo estimation for $E_A(\boldsymbol{P}_{AB}^\pi)$. Notice Eq. (9) is *convex* for $\boldsymbol{P}_A$, which enables efficient solutions. As a result, Eq. (9) provides an efficient estimation of Markov entanglement via simulation and can be easily extend to $N$-agent MDPs.

**Numerical experiments.** Finally, we empirically study the value decomposition for the index policy on a circulant RMAB benchmark [3, 52, 10, 18] that has 4 different states each local agent. As a result, the global state space scales as large as $4^{1800} > 10^{1000}$ for $N = 1800$ agents. The specific transitions and rewards are introduced in Appendix K. For each RMAB instance, we sample a trajectory of length $T = 5N$ and use the collected data to i) solve Eq. (9) to estimate the measure of Markov entanglement; ii) train local Q-value decomposition. It quickly follows from the results in Figure 1:

The estimated Markov entanglement decays as $\mathcal{O}(1/\sqrt{N})$ in the left panel, consistent with theoretical predictions. This also implies a low decomposition error scaling of $\mathcal{O}(\sqrt{N})$, as seen in the right panel. Furthermore, the simulated trajectory has a length of $T = 5N$ while the global state space has size $|S|^N$, making both entanglement estimation and local Q-value decomposition sample-efficient.

## 6 Discussions

**Comparison with quantum entanglement** One notable difference between the definition of Markov and quantum entanglement is that the former does not require coefficients $\boldsymbol{x} \ge 0$. In Appendix C, we show there exist separable two-agent MDPs that can only be represented by linear combinations but not convex combinations of independent subsystems, highlighting a structural difference between Markov and quantum entanglement. Finally, we emphasize that our analogy to quantum entanglement is mostly in the mathematical formulation; there is no clear physical interpretation analogy between Markov and quantum entanglement.

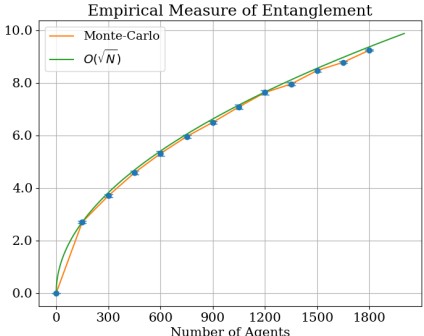
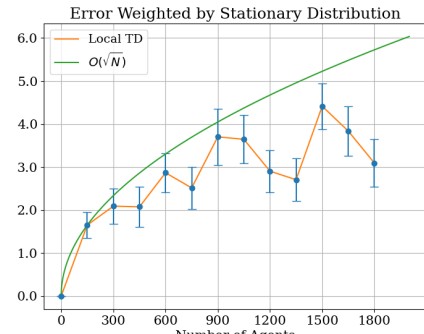

Figure 1: Circulant RMAB under an index policy. *Left:* empirical estimation of Markov entanglement multiplied by the number of agents, $NE_1(\boldsymbol{P}_{1:N}^{\pi})$. *Right:* $\mu$-weighted decomposition error.

**Relations to Influenced-based MARL**   There's another line of MARL research that explicitly models the influence of other agents as intrinsic rewards for exploration [45, 26]. It turns out the mutual information can be viewed as the measure of Markov entanglement under KL-divergence. Specifically, we can rewrite mutual information in [45] as

$$I(S_2', A_2'; S_2, A_2|S_1, A_1) = \sum_{\boldsymbol{s}, \boldsymbol{a}, s_2', a_2'} p^{\pi}(\boldsymbol{s}, \boldsymbol{a}, s_2', a_2') \left( \log \frac{p^{\pi}(s_2', a_2'|\boldsymbol{s}, \boldsymbol{a})}{p^{\pi}(s_2', a_2'|s_2, a_2)} \right)$$
$$= \sum_{\boldsymbol{s}, \boldsymbol{a}} \mu^{\pi}(\boldsymbol{s}, \boldsymbol{a}) D_{KL}\left(p^{\pi}(\cdot|\boldsymbol{s}, \boldsymbol{a})||P_2(\cdot|s_2, a_2)\right) .$$

This is highly related to our measure of Markov entanglement under a $\mu^{\pi}$-weighted agent-wise KL-divergence, which we can define as

$$E_2(\boldsymbol{P}_{12}) = \min_{\boldsymbol{P}_2} \sum_{\boldsymbol{s}, \boldsymbol{a}} \mu^{\pi}(\boldsymbol{s}, \boldsymbol{a}) D_{KL}\left(p^{\pi}(\cdot|\boldsymbol{s}, \boldsymbol{a})||P_2(\cdot|s_2, a_2)\right) .$$

Intuitively, the measure of Markov entanglement can be viewed as how closely one agent can be approximated as an independent subsystem. This characterization aligns naturally with mutual information. Furthermore, since KL-divergence provides an upper bound for total variation distance, it consequently bounds our Markov entanglement measure relative to the ATV distance introduced in our paper. This connection demonstrates that influence-based MARL methods naturally fit within our theoretical framework, corresponding to a specialized distance measure.

# 7   Conclusion

This paper established the mathematical foundation of value decomposition in MARL. Drawing inspiration from quantum physics, we propose the idea of Markov entanglement and prove that it serves as a sufficient and necessary condition for the exact value decomposition. We further characterize the decomposition error in general multi-agent MDPs through the measure of Markov entanglement. As application examples, we prove widely-used index policies are asymptotically separable and suggest practitioners using Markov entanglement as a proxy for estimating the effectiveness of value decomposition.

## Acknowledgments and Disclosure of Funding

We thank all anonymous reviewers for their constructive comments. We are also grateful to Prof. Tongyang Li for valuable and insightful discussions.

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

# Contents

## A  Linear algebra with tensor product

We briefly introduce the basic properties of tensor product or Kronecker product. Let $\boldsymbol{A} \in \mathbb{R}^{m_1 \times n_1}, \boldsymbol{B} \in \mathbb{R}^{m_2 \times n_2}$, then

$$
\boldsymbol{A} \otimes \boldsymbol{B} = \left[ \begin{array}{cccc} a_{11}\boldsymbol{B} & a_{12}\boldsymbol{B} & \ldots & a_{1n_1}\boldsymbol{B} \\ a_{21}\boldsymbol{B} & a_{22}\boldsymbol{B} & \ldots & a_{2n_1}\boldsymbol{B} \\ \ldots & \ldots & \ldots & \ldots \\ a_{m_1 1}\boldsymbol{B} & a_{m_1 2}\boldsymbol{B} & \ldots & a_{m_1 n_1}\boldsymbol{B} \end{array} \right] \in \mathbb{R}^{m_1 m_2 \times n_1 n_2} \, .
$$

Tensor product satisfies the following basic properties,

- **1. Bilinearity** For any matrix $\boldsymbol{A}, \boldsymbol{B}, \boldsymbol{C}$ and constant $k$, it holds $k(\boldsymbol{A} \otimes \boldsymbol{B}) = (k\boldsymbol{A}) \otimes \boldsymbol{B} = \boldsymbol{A} \otimes (k\boldsymbol{B})$, $(\boldsymbol{A} + \boldsymbol{B}) \otimes \boldsymbol{C} = \boldsymbol{A} \otimes \boldsymbol{C} + \boldsymbol{B} \otimes \boldsymbol{C}$, and $\boldsymbol{A} \otimes (\boldsymbol{B} + \boldsymbol{C}) = \boldsymbol{A} \otimes \boldsymbol{B} + \boldsymbol{A} \otimes \boldsymbol{C}$.
- **2. Mixed-product Property** For any matrix $\boldsymbol{A}, \boldsymbol{B}, \boldsymbol{C}, \boldsymbol{D}$, if $\boldsymbol{AC}$ and $\boldsymbol{BD}$ form valid matrix product, then $(\boldsymbol{A} \otimes \boldsymbol{B})(\boldsymbol{C} \otimes \boldsymbol{D}) = (\boldsymbol{AC}) \otimes (\boldsymbol{BD})$.

## B  Decompose value functions

Compared to the decomposition of Q-value, the value function further requires the reward to be *state-dependent*. To illustrate, notice by Bellman equation,

$$
V_{AB}^{\pi} = (\boldsymbol{I} - \gamma \boldsymbol{P}_{AB}^{\pi})^{-1} \boldsymbol{r}_{AB}^{\pi} \, ,
$$

where we abuse notation and denote $P_{AB}^{\pi}(\boldsymbol{s}' \mid \boldsymbol{s}) = \sum_{\boldsymbol{a}} \pi(\boldsymbol{a} \mid \boldsymbol{s}) P(\boldsymbol{s}' \mid \boldsymbol{s}, \boldsymbol{a})$ and reward $r_{AB}^{\pi}(\boldsymbol{s}) = \sum_{\boldsymbol{a}} \pi(\boldsymbol{a} \mid \boldsymbol{s}) r_{AB}(\boldsymbol{s}, \boldsymbol{a})$. A key subtlety arises because $\boldsymbol{r}_{AB}^{\pi}$ may not be decomposable—even when $\boldsymbol{r}_{AB}$ is decomposable—unless the reward $\boldsymbol{r}_{AB}$ is state-dependent. Consequently, we cannot directly apply the "absorbing" equation as in the proof of Theorem 1.

On the other hand, Q-value decomposition bypasses the state-dependence assumption and provides a stronger condition that directly implies value function decomposition. As a result, while learning local value functions may seem more intuitive, we recommend learning local Q-values instead and using them to approximate the global value function.

## C  Comparison with quantum entanglement

It turns out that our Markov entanglement condition serves as a mathematical counterpart of quantum entanglement in quantum physics. We provide the formal definition of the latter for comparison.

**Definition 4** (Two-party Quantum Entanglement)**.** *Consider a two-party quantum system composed of two subsystems $A$ and $B$. The joint state $\rho_{AB}$ is **separable** if there exists $K \in \mathbb{Z}^+$, a probability measure $\{x_j\}_{j \in [K]}$, and density matrices $\left\{ \rho_A^{(j)}, \rho_B^{(j)} \right\}_{j \in [K]}$ such that*

$$
\rho_{AB} = \sum_{j=1}^{K} x_j \rho_A^{(j)} \otimes \rho_B^{(j)} \, .
$$

*If there exists no such decomposition, $\rho_{AB}$ is **entangled**.*

The density matrices are square matrices satisfying certain properties such as positive semi-definiteness and trace normalization, which can be viewed as the counterparts of transition matrices

in the Markov world. Despite the similarities in mathematical form, quantum entanglement imposes an additional constraint requiring $\{x_j\}_{j\in[K]}$ to be a probability measure, i.e. $\boldsymbol{x} \geq 0$. In contrast, our Markov entanglement defined in Definition 1 permits general linear coefficients $\{x_j\}_{j\in[K]}$ as long as $\sum_{j=1}^{k} x_j = 1$. This distinction raises the important question of whether negative coefficients are indeed necessary in characterizing Markov entanglement.

To start with, we introduce the set of all separable transition matrices

$$\mathcal{P}_{\text{SEP}} = \left\{ \boldsymbol{P} \geq 0 \;\middle|\; \boldsymbol{P} = \sum_{j=1}^{K} x_j \boldsymbol{P}_A^{(j)} \otimes \boldsymbol{P}_B^{(j)} \,,\; \sum_{j=1}^{K} x_j = 1 \right\} \,,$$

where $K \in \mathbb{Z}^+$ and $\left\{ \boldsymbol{P}_A^{(j)}, \boldsymbol{P}_B^{(j)} \right\}_{j\in[K]}$ are transition matrices. $\boldsymbol{P} \geq 0$ calls for every element of $\mathcal{P}_{\text{SEP}}$ to be a valid transition matrix. It's clear that a transition matrix $\boldsymbol{P}_{AB}^{\pi}$ is separable if and only if $\boldsymbol{P}_{AB}^{\pi} \in \mathcal{P}_{\text{SEP}}$. On the other hand, a direct analogy of quantum entanglement gives us the following set that further requires non-negative coefficients,

$$\mathcal{P}_{\text{SEP}}^+ = \left\{ \boldsymbol{P} \geq 0 \;\middle|\; \boldsymbol{P} = \sum_{j=1}^{K} x_j \boldsymbol{P}_A^{(j)} \otimes \boldsymbol{P}_B^{(j)} \,,\; \sum_{j=1}^{K} x_j = 1 \,,\; \boldsymbol{x} \geq 0 \right\} \,.$$

Interestingly, it turns out $\mathcal{P}_{\text{SEP}}^+ \not\subseteq \mathcal{P}_{\text{SEP}}$. In other words, there exist separable two-agent MDPs that can only be represented by linear combinations but not convex combinations of independent subsystems. Specifically, consider the following basis

$$\boldsymbol{E}_{00} = \begin{pmatrix} 1 & 0 \\ 1 & 0 \end{pmatrix}, \quad \boldsymbol{E}_{01} = \begin{pmatrix} 1 & 0 \\ 0 & 1 \end{pmatrix}, \quad \boldsymbol{E}_{10} = \begin{pmatrix} 0 & 1 \\ 1 & 0 \end{pmatrix}, \quad \boldsymbol{E}_{11} = \begin{pmatrix} 0 & 1 \\ 0 & 1 \end{pmatrix}$$

And the corresponding transition matrix we provide is

$$\boldsymbol{P} = \begin{pmatrix} 0.5 & 0 & 0 & 0.5 \\ 0.5 & 0 & 0 & 0.5 \\ 0.5 & 0 & 0 & 0.5 \\ 0 & 0.5 & 0.5 & 0 \end{pmatrix} = \frac{1}{2}\boldsymbol{E}_{00} \otimes \boldsymbol{E}_{00} + \frac{1}{2}\boldsymbol{E}_{10} \otimes \boldsymbol{E}_{11} + \frac{1}{2}\boldsymbol{E}_{11} \otimes \boldsymbol{E}_{10} - \frac{1}{2}\boldsymbol{E}_{10} \otimes \boldsymbol{E}_{10}$$

One can also verify $\boldsymbol{P}$ can not be represented by the convex combination of tensor products of these basis. This result justifies the necessity of negative coefficients in $\boldsymbol{x}$ and highlights a structural difference between Markov entanglement and quantum entanglement

# D   Proof of Theorem 2

We provide the full proof of Theorem 2 in this section.

**Step 1: Characterize the Orthogonal Complement.**   To start with, we consider the smallest subspace containing all transition matrices $\Omega_P := \text{span}(\text{P})$ where $\boldsymbol{P}$ are the set of all transition matrices in $\mathbb{R}^{m \times m}$. We then study the dimension of $\Omega_P$.

**Lemma 1.** *The dimension of $\Omega_P$ is $\dim(\Omega_\text{P}) = \text{m}^2 - \text{m} + 1$.*

*Proof.* Let $\boldsymbol{Z}_{ij} \in \mathbb{R}^{m \times m}$ such that

$$\boldsymbol{Z}_{ij}(a,b) = \begin{cases} 1 & (a = i \wedge b = j) \vee (a = b) \\ 0 & o.w. \end{cases} \,.$$

One basis for all transition matrices is given by $\{\boldsymbol{Z}_{ij}\}_{i,j\in[m]}$ whose cardinarlity is $m^2 - m + 1$.   $\square$

Let $\Omega_{P\otimes 2} := \text{span}(\text{P}_1 \otimes \text{P}_2)$ be the minimal subspace containing all separable transition matrices. It quickly follows that
$$\dim(\Omega_{\text{P}\otimes 2}) = (\dim(\Omega_\text{P}))^2 \,.$$

We then construct the orthogonal complement of $\Omega_{P^{\otimes 2}}$ under Frobenius inner product. Let $\{\varepsilon_j\}_{j\in[m-1]}$ be a set of vector in $\mathbb{R}^m$ such that $\varepsilon_j = (1, 0, \ldots, 0, -1, 0, \ldots, 0)^\top$ with the first element 1 and $j+1$-th element $-1$. Notice that

$$\mathrm{Tr}\left(e\varepsilon_j^\top P\right) = \mathrm{Tr}\left(\varepsilon_j^\top P e\right) = 0\,,$$

for all $\varepsilon_j$. Consider the following subspace

$$\Omega' = \left\{ \sum_{j=1}^{m-1} \left(\varepsilon_j e^\top\right) \otimes W_j^1 + \sum_{j=1}^{m-1} W_j^2 \otimes \left(\varepsilon_j e^\top\right) \mid W_{1:j}^1, W_{1:j}^2 \in \mathbb{R}^{m\times m} \right\}\,.$$

We then show $\Omega'$ is exactly the orthogonal complement of $\Omega_{P^{\otimes 2}}$. First, notice that

$$\dim(\Omega') = 2(\mathrm{m}-1)\mathrm{m}^2 - (\mathrm{m}-1)^2\,.$$

and thus $\dim(\Omega') + \dim(\Omega_{P^{\otimes 2}}) = \mathrm{m}^4$. Moreover, one can verify for any $X \in \Omega_{P^{\otimes 2}}$ and $Y \in \Omega'$, $\mathrm{Tr}(X^\top Y) = 0$. As a result, it holds

$$\Omega' = \Omega_{P^{\otimes 2}}^\perp\,.$$

**Step 2: Connection to "Inverse"** The decomposition of Q-value ultimately concerns with the properties of $(I - \gamma P_{AB}^\pi)^{-1}$. The following lemma bridges this gap.

**Lemma 2.** *Given any transition matrix $P$ and $\gamma > 0$, $P$ is separable if and only if $(1-\gamma)(I-\gamma P)^{-1}$ is separable.*

*Proof.* ($\Rightarrow$) One can verify that $(I - \gamma P)e = (1-\gamma)e$, which implies $(1-\gamma)(I-\gamma P)^{-1}$ is a transition matrix. Moreover, $(1-\gamma)(I-\gamma P)^{-1} = (1-\gamma)\sum_{i=0}^\infty (\gamma P)^i$ falls in $\Omega_{P^{\otimes 2}}$ as $P \in \Omega_{P^{\otimes 2}}$.

($\Leftarrow$) This side is more involved. Denote $U := (1-\gamma)(I-\gamma P)^{-1}$. Then if the spectral radius $\rho(I - U) < 1$, then

$$U^{-1} = (I - (I - U))^{-1} = \sum_{i=0}^\infty (I - U)^i \in \Omega_{P^{\otimes 2}}\,.$$

This implies $U^{-1} = \frac{1}{1-\gamma}(I - \gamma P) \in \Omega_{P^{\otimes 2}}$ and thus $P \in \Omega_{P^{\otimes 2}}$, finishing the proof. It then suffices to show $\rho(I - U) < 1$. Notice that

$$\lambda_i(I - U) = 1 - \lambda_i(U) = 1 - \frac{1-\gamma}{\lambda(I - \gamma P)} = 1 - \frac{1-\gamma}{1 - \gamma\lambda_i(P)}\,.$$

Let $\lambda_i(P) = a + bi$ and taking modulus for both side

$$\begin{aligned}
|\lambda_i(I - U)| &= \left|\frac{\gamma - \gamma\lambda_i(P)}{1 - \gamma\lambda_i(P)}\right| \\
&= \frac{|\gamma - \gamma\lambda_i(P)|}{|1 - \gamma\lambda_i(P)|} \\
&= \sqrt{\frac{\gamma^2(1-a)^2 + \gamma^2 b^2}{(1-\gamma a)^2 + \gamma^2 b^2}} \\
&= \sqrt{1 + \frac{(1-\gamma)(2a\gamma - \gamma - 1)}{(1-\gamma a)^2 + \gamma^2 b^2}} \\
&\leq \sqrt{1 - \frac{(1-\gamma)^2}{(1-\gamma a)^2 + \gamma^2 b^2}} < 1\,.
\end{aligned}$$

We conclude the proof given $\rho(I - U) = \max_i |\lambda_i(I - U)| < 1$. $\qquad\square$

**Step 3: Put it together** By Lemma 2, if $\boldsymbol{P}_{AB}^\pi$ is entangled, then $(1-\gamma)(\boldsymbol{I}-\gamma\boldsymbol{P}_{AB}^\pi)^{-1}$ is also entangled. Then there exists $\boldsymbol{Y}\in\Omega'\neq\boldsymbol{0}$ such that $\mathrm{Tr}(\boldsymbol{Y}^\top(\boldsymbol{I}-\gamma\boldsymbol{P}_{AB}^\pi)^{-1})\neq 0$. We apply singular value decomposition to all $W_{1:j}^1, W_{1:j}^2$ and conclude there exists some $j$ and $\boldsymbol{u},\boldsymbol{v}\in\mathbb{R}^m$ such that either $\mathrm{Tr}((\boldsymbol{e}\varepsilon_j^\top)\otimes(\boldsymbol{v}\boldsymbol{u}^\top)(\boldsymbol{I}-\gamma\boldsymbol{P}_{AB}^\pi)^{-1})\neq 0$ or $\mathrm{Tr}((\boldsymbol{v}\boldsymbol{u}^\top)\otimes(\boldsymbol{e}\varepsilon_j^\top)(\boldsymbol{I}-\gamma\boldsymbol{P}_{AB}^\pi)^{-1})\neq 0$. We assume the former without loss of generality, it holds

$$(\varepsilon_j^\top\otimes\boldsymbol{u}^\top)(\boldsymbol{I}-\gamma\boldsymbol{P}_{AB}^\pi)^{-1}(\boldsymbol{e}\otimes\boldsymbol{v})\neq 0\,.$$

Now set $\boldsymbol{r}_A=\boldsymbol{0}$ and $\boldsymbol{r}_B=\boldsymbol{v}$. Since $Q_{AB}^\pi$ is decomposable, there exists some local function $Q_A, Q_B$ such that

$$(\boldsymbol{I}-\gamma\boldsymbol{P}_{AB}^\pi)^{-1}(\boldsymbol{e}\otimes\boldsymbol{v})=Q_A(\boldsymbol{0})\otimes\boldsymbol{e}+\boldsymbol{e}\otimes Q_B(\boldsymbol{v})\,.$$

Left multiply by $(\varepsilon_j^\top\otimes\boldsymbol{u}^\top)$, we have

$$(\varepsilon_j^\top\otimes\boldsymbol{u}^\top)(\boldsymbol{I}-\gamma\boldsymbol{P}_{AB}^\pi)^{-1}(\boldsymbol{e}\otimes\boldsymbol{v})=(\varepsilon_j^\top\otimes\boldsymbol{u}^\top)(Q_A(\boldsymbol{0})\otimes\boldsymbol{e})\neq 0\,,$$

Then set $\boldsymbol{r}_A=\boldsymbol{0}$ and $\boldsymbol{r}_B=-\boldsymbol{v}$, we can similarly derive

$$-(\varepsilon_j^\top\otimes\boldsymbol{u}^\top)(\boldsymbol{I}-\gamma\boldsymbol{P}_{AB}^\pi)^{-1}(\boldsymbol{e}\otimes\boldsymbol{v})=(\varepsilon_j^\top\otimes\boldsymbol{u}^\top)(Q_A(\boldsymbol{0})\otimes\boldsymbol{e})\neq 0\,,$$

This gives use $(\varepsilon_j^\top\otimes\boldsymbol{u}^\top)(Q_A(\boldsymbol{0})\otimes\boldsymbol{e})=0$, which is a contradiction.

# E   Decomposition via general functions

Entangled $\boldsymbol{P}$ precludes the local decomposition with local value functions, but may admit decompositions with more general functions. Consider $\boldsymbol{P}=\frac{1}{4}\left(ee^\top\right)\otimes\left(ee^\top\right)+\delta\left(\epsilon\epsilon^\top\right)\otimes\left(\epsilon\epsilon^\top\right)$, where $e=[1,1], \epsilon=[1-1]$. Clearly such $\boldsymbol{P}$ is entangled. We also have $\boldsymbol{P}^k=\frac{1}{4}\left(ee^\top\right)\otimes\left(ee^\top\right)$ for $k\geq 2$. Then $(I-\gamma P)^{-1}=\boldsymbol{I}+\frac{\gamma+\gamma^2}{4}\left(ee^\top\right)\otimes\left(ee^\top\right)+\delta\gamma\left(\epsilon\epsilon^\top\right)\otimes\left(\epsilon\epsilon^\top\right)$. Then for any $\boldsymbol{r}_A,\boldsymbol{r}_B$, we have $(\boldsymbol{I}-\gamma\boldsymbol{P})^{-1}\left(\boldsymbol{r}_A\otimes e+e\otimes\boldsymbol{r}_B\right)=\boldsymbol{r}_A\otimes e+h_A\left(\gamma+\gamma^2\right)/2e\otimes e+\boldsymbol{r}_B\otimes e+h_B\left(\gamma+\gamma^2\right)/2e\otimes e+2\delta\gamma\left(\epsilon^\top\boldsymbol{r}_B\right)\epsilon\otimes e$ where $h_A=e^\top\boldsymbol{r}_A, h_B=e^\top\boldsymbol{r}_B$.

# F   Proof of Theorem 3

**Additional Notations** For (semi-)norm $\|\cdot\|_\alpha$ and norm $\|\cdot\|_\beta$, we define the $\alpha,\beta$-norm for matrix $\boldsymbol{A}$ as

$$\|\boldsymbol{A}\|_{\alpha,\beta}=\sup_{\|\boldsymbol{x}\|_\beta=1}\|\boldsymbol{A}\boldsymbol{x}\|_\alpha\,.$$

We further abbreviate $\|\boldsymbol{A}\|_\alpha:=\|\boldsymbol{A}\|_{\alpha,\alpha}$. Moreover, we define the operator $|\boldsymbol{x}|$ taking the absolute value of each element of vector or matrix $\boldsymbol{x}$.

To prove the theorem, we introduce the key technique of analyzing perturbation bounds of the transition matrix, which is also used in [17].

**Lemma 3** (Lemma 1 in [17]). *Let $\boldsymbol{P},\boldsymbol{P}'\in\mathbb{R}^{n\times n}$ such that $(\boldsymbol{I}-\boldsymbol{P})^{-1}$ and $(\boldsymbol{I}-\boldsymbol{P}')^{-1}$ exist. Then it holds*

$$(\boldsymbol{I}-\boldsymbol{P}')^{-1}=(\boldsymbol{I}-\boldsymbol{P})^{-1}+(\boldsymbol{I}-\boldsymbol{P}')^{-1}(\boldsymbol{P}'-\boldsymbol{P})(\boldsymbol{I}-\boldsymbol{P})^{-1}\,.$$

We are then ready to prove the main theorem.

*Proof of Theorem 3.* Let $\boldsymbol{P}_A, \boldsymbol{P}_B$ be the optimal solution to Eq. (7) w.r.t agent $A, B$. For any subset of state-action pairs of agent $A$, $\mathcal{F} \subseteq \mathcal{S}_A \times \mathcal{A}_A$, we have

$$
\left| \sum_{s'_A, a'_A \in \mathcal{F}} (\boldsymbol{P}_A^\pi - \boldsymbol{P}_A)_{(s'_A, a'_A \mid s_A, a_A)} \right|
$$

$$
= \left| \sum_{s'_A, a'_A \in \mathcal{F}} \sum_{s'_B, a'_B} \sum_{s_B, a_B} (\boldsymbol{P}_{AB}^\pi - \boldsymbol{P}_A \otimes \boldsymbol{P}_B)_{(\boldsymbol{s}', \boldsymbol{a}' \mid \boldsymbol{s}, \boldsymbol{a})} \, \mu_{AB}^\pi(s_B, a_B \mid s_A, a_A) \right|
$$

$$
\leq \sum_{s_B, a_B} \left| \sum_{s'_A, a'_A \in \mathcal{F}} \sum_{s'_B, a'_B} (\boldsymbol{P}_{AB}^\pi - \boldsymbol{P}_A \otimes \boldsymbol{P}_B)_{(\boldsymbol{s}', \boldsymbol{a}' \mid \boldsymbol{s}, \boldsymbol{a})} \right| \mu_{AB}^\pi(s_B, a_B \mid s_A, a_A)
$$

$$
\leq \sum_{s_B, a_B} E_A(\boldsymbol{P}_{AB}^\pi) \mu_{AB}^\pi(s_B, a_B \mid s_A, a_A) = E_A(\boldsymbol{P}_{AB}^\pi)
$$

where the last inequality follows from the definition of agent-wise total variation distance. Since the result holds for any $\mathcal{F}$ and $(s_A, a_A) \in \mathcal{S}_A \times \mathcal{A}_A$, we have

$$
\|\boldsymbol{P}_A^\pi - \boldsymbol{P}_A\|_{\mathrm{TV}} \leq E_A(\boldsymbol{P}_{AB}^\pi),
$$

and similar results hold for $\boldsymbol{P}_B^\pi$.

Next we have

$$
(\boldsymbol{I} - \gamma \boldsymbol{P}_{AB}^\pi)^{-1} (\boldsymbol{r}_A \otimes \boldsymbol{e}) - \left( (\boldsymbol{I} - \gamma \boldsymbol{P}_A^\pi)^{-1} \boldsymbol{r}_A \right) \otimes \boldsymbol{e}
$$

$$
= (\boldsymbol{I} - \gamma \boldsymbol{P}_{AB}^\pi)^{-1} (\boldsymbol{r}_A \otimes \boldsymbol{e}) - (\boldsymbol{I} - \gamma \boldsymbol{P}_A \otimes \boldsymbol{P}_B)^{-1} (\boldsymbol{r}_A \otimes \boldsymbol{e})
$$

$$
+ (\boldsymbol{I} - \gamma \boldsymbol{P}_A \otimes \boldsymbol{P}_B)^{-1} (\boldsymbol{r}_A \otimes \boldsymbol{e}) - \left( (\boldsymbol{I} - \gamma \boldsymbol{P}_A^\pi)^{-1} \boldsymbol{r}_A \right) \otimes \boldsymbol{e}
$$

$$
\stackrel{(i)}{=} \underbrace{(\boldsymbol{I} - \gamma \boldsymbol{P}_{AB}^\pi)^{-1} (\boldsymbol{r}_A \otimes \boldsymbol{e}) - (\boldsymbol{I} - \gamma \boldsymbol{P}_A \otimes \boldsymbol{P}_B)^{-1} (\boldsymbol{r}_A \otimes \boldsymbol{e})}_{(I)}
$$

$$
+ \underbrace{\left( (\boldsymbol{I} - \gamma \boldsymbol{P}_A)^{-1} \boldsymbol{r}_A \right) \otimes \boldsymbol{e} - \left( (\boldsymbol{I} - \gamma \boldsymbol{P}_A^\pi)^{-1} \boldsymbol{r}_A \right) \otimes \boldsymbol{e}}_{(II)}
$$

where $(i)$ also follows the same "absorbing" technique in the proof of Theorem 1.

For $(I)$, apply Lemma 3, it holds

$$
\left\| (\boldsymbol{I} - \gamma \boldsymbol{P}_{AB}^\pi)^{-1} (\boldsymbol{r}_A \otimes \boldsymbol{e}) - (\boldsymbol{I} - \gamma \boldsymbol{P}_A \otimes \boldsymbol{P}_B)^{-1} (\boldsymbol{r}_A \otimes \boldsymbol{e}) \right\|_\infty
$$

$$
= \left\| (\boldsymbol{I} - \gamma \boldsymbol{P}_{AB}^\pi)^{-1} (\gamma \boldsymbol{P}_{AB}^\pi - \gamma \boldsymbol{P}_A \otimes \boldsymbol{P}_B) (\boldsymbol{I} - \gamma \boldsymbol{P}_A \otimes \boldsymbol{P}_B)^{-1} (\boldsymbol{r}_A \otimes \boldsymbol{e}) \right\|_\infty
$$

$$
\leq \left\| (\boldsymbol{I} - \gamma \boldsymbol{P}_{AB}^\pi)^{-1} \right\|_\infty \left\| (\gamma \boldsymbol{P}_{AB}^\pi - \gamma \boldsymbol{P}_A \otimes \boldsymbol{P}_B) \left( (\boldsymbol{I} - \gamma \boldsymbol{P}_A)^{-1} \boldsymbol{r}_A \right) \otimes \boldsymbol{e} \right\|_\infty
$$

$$
\stackrel{(i)}{\leq} \left\| (\boldsymbol{I} - \gamma \boldsymbol{P}_{AB}^\pi)^{-1} \right\|_\infty 2\gamma E_A(\boldsymbol{P}_{AB}^\pi) \left\| (\boldsymbol{I} - \gamma \boldsymbol{P}_A)^{-1} \boldsymbol{r}_A \right\|_\infty
$$

$$
\leq \frac{2\gamma E_A(\boldsymbol{P}_{AB}^\pi) r_{\max}^A}{1 - \gamma} \left\| (\boldsymbol{I} - \gamma \boldsymbol{P}_{AB}^\pi)^{-1} \right\|_\infty \leq \frac{2\gamma E_A(\boldsymbol{P}_{AB}^\pi) r_{\max}^A}{(1 - \gamma)^2},
$$

where $(i)$ follows by the definition of agent-wise total variation distance when $\|\boldsymbol{r}_A\|_\infty \neq 0$, and also trivially hold when $\|\boldsymbol{r}_A\|_\infty = 0$. Similarly, for $(II)$ we have

$$
\left\| \left( (\boldsymbol{I} - \gamma \boldsymbol{P}_A)^{-1} \boldsymbol{r}_A \right) \otimes \boldsymbol{e} - \left( (\boldsymbol{I} - \gamma \boldsymbol{P}_A^\pi)^{-1} \boldsymbol{r}_A \right) \otimes \boldsymbol{e} \right\|_\infty
$$

$$
= \left\| \left( (\boldsymbol{I} - \gamma \boldsymbol{P}_A)^{-1} - (\boldsymbol{I} - \gamma \boldsymbol{P}_A^\pi)^{-1} \right) \boldsymbol{r}_A \right\|_\infty
$$

$$
= \left\| (\boldsymbol{I} - \gamma \boldsymbol{P}_A^\pi)^{-1} (\gamma \boldsymbol{P}_A^\pi - \gamma \boldsymbol{P}_A) (\boldsymbol{I} - \gamma \boldsymbol{P}_A)^{-1} \boldsymbol{r}_A \right\|_\infty
$$

$$
\leq \frac{2\gamma E_A(\boldsymbol{P}_{AB}^\pi) r_{\max}^A}{(1-\gamma)^2} .
$$

Then we have

$$
\left\| (\boldsymbol{I} - \gamma \boldsymbol{P}_{AB}^\pi)^{-1} (\boldsymbol{r}_A \otimes \boldsymbol{e}) - \left( (\boldsymbol{I} - \gamma \boldsymbol{P}_A^\pi)^{-1} \boldsymbol{r}_A \right) \otimes \boldsymbol{e} \right\|_\infty \leq \frac{4\gamma E_A(\boldsymbol{P}_{AB}^\pi) r_{\max}^A}{(1-\gamma)^2} .
$$

We can derive similar results for agent $B$, i.e.,

$$
\left\| (\boldsymbol{I} - \gamma \boldsymbol{P}_{AB}^\pi)^{-1} (\boldsymbol{e} \otimes \boldsymbol{r}_B) - \boldsymbol{e} \otimes \left( (\boldsymbol{I} - \gamma \boldsymbol{P}_B^\pi)^{-1} \boldsymbol{r}_B \right) \right\|_\infty \leq \frac{4\gamma E_B(\boldsymbol{P}_{AB}^\pi) r_{\max}^B}{(1-\gamma)^2} .
$$

Put it all together we have

$$
\left\| Q_{AB}^\pi - (Q_A^\pi \otimes \boldsymbol{e} + \boldsymbol{e} \otimes Q_B^\pi) \right\|_\infty \leq \frac{4\gamma (E_A(\boldsymbol{P}_{AB}^\pi) r_{\max}^A + E_B(\boldsymbol{P}_{AB}^\pi) r_{\max}^B)}{(1-\gamma)^2} .
$$

$\square$

# G   Proof of Theorem 4

We first introduce the $\mu$-weighted ATV distance Formally, we introduce the following norm.

**Definition 5** ($\mu$-norm). *Given a transition matrix $\boldsymbol{P} \in \mathbb{R}^{|\mathcal{S}||\mathcal{A}| \times |\mathcal{S}||\mathcal{A}|}$ with occupancy measure[5] $\mu \in \mathbb{R}^{|\mathcal{S}||\mathcal{A}|}$, for any vector $\boldsymbol{x} \in \mathbb{R}^{|\mathcal{S}||\mathcal{A}|}$ the $\mu$-norm is defined as*

$$
\|\boldsymbol{x}\|_\mu := \sum_{(s,a) \in \mathcal{S} \times \mathcal{A}} \mu(s,a) |x(s,a)| = \mu^\top |\boldsymbol{x}| . \tag{10}
$$

One can verify that $\mu$-norm satisfies triangle inequality and is a valid norm when $\mu(s,a) > 0$ for all $(s,a)$. Otherwise $\mu$-norm is a *semi-norm* in general. We then introduce the distance

**Definition 6** ($\mu$-weighted Agent-wise Total Variation Distance). *Given probability distribution $\mu \in \mathbb{R}^{|S|^2|A|^2}$, the $\mu$-weighted total variation distance between two transition matrices $\boldsymbol{P}, \boldsymbol{P}' \in \mathbb{R}^{|S|^2|A|^2 \times |S|^2|A|^2}$ w.r.t agent $A$ is defined as*

$$
\|\boldsymbol{P} - \boldsymbol{P}'\|_{\mu-\mathrm{ATV}_A} = \frac{1}{2} \sup_{\|\boldsymbol{x}\|_\infty = 1} \| (\boldsymbol{P} - \boldsymbol{P}') (\boldsymbol{x} \otimes \boldsymbol{e}) \|_\mu .
$$

The $\mu$-weighted ATV distance w.r.t agent $B$ can be defined similarly. We claim that the $\mu$-weighted ATV is also a counterpart of ATV distance in Definition 6. This follows from the constrained optimization formulation of ATV

$$
\|\boldsymbol{P} - \boldsymbol{P}'\|_{\mathrm{ATV}_A} = \frac{1}{2} \sup_{\|\boldsymbol{x}\|_\infty = 1} \| (\boldsymbol{P} - \boldsymbol{P}') (\boldsymbol{x} \otimes \boldsymbol{e}) \|_\infty . \tag{11}
$$

Thus $\mu$-ATV substitutes $\mu$-norm for the original $\ell_\infty$-norm. We plug $\mu$-weighted ATV into Eq. (1) and obtain the corresponding measure of Markov entanglement $E(\boldsymbol{P}_{AB}^\pi)$ and $E_A(\boldsymbol{P}_{AB}^\pi)$. Similar to ATV in Eq. (7), this $\mu$-weighted version of $E_A(\boldsymbol{P}_{AB}^\pi)$ admits the following formulation

$$
E_A(\boldsymbol{P}_{AB}^\pi) \leq \min_{\boldsymbol{P}_A} \sum_{s,a} \rho_{AB}^\pi(s,a) D_{\mathrm{TV}} \left( \boldsymbol{P}_{AB}^\pi(\cdot,\cdot \mid s,a), \boldsymbol{P}_A(\cdot,\cdot \mid s_A, a_A) \right) . \tag{12}
$$

---

[5]Since $\mu \in \mathbb{R}^{|\mathcal{S}||\mathcal{A}|}$ is the stationary distribution of $\boldsymbol{P} \in \mathbb{R}^{|\mathcal{S}||\mathcal{A}| \times |\mathcal{S}||\mathcal{A}|}$, we use "stationary distribution" and "occupancy measure" exchangeably when the context is clear.

This recovers Eq. (8) that substitutes the $\mu$-weighted average for the maximum operator in Eq. (7). Thus intuitively, $E(\boldsymbol{P}_{AB}^\pi)$ w.r.t $\mu$-weighted ATV distance measures *how closely agent A can be approximated as an independent subsystem under the stationary distribution.*

We provide the proof for two agents here, one can easily generalize the proof to multi-agent scenarios. Compared to the proof of Theorem 3, this proof follows similar framework and differs in several details.

The first one is the following lemma for the "localized" stationary distribution

**Lemma 4.** $\boldsymbol{P}_A^\pi$ *has stationary distribution* $\mu_A^\pi$ *with*

$$\forall (s_A, a_A) , \, \mu_A^\pi(s_A, a_A) = \sum_{s_B, a_B} \mu_{AB}^\pi(s_A, s_B, a_A, a_B) \,.$$

In other words, the local stationary distribution of each agent is exactly the marginal distribution of global $\mu_{AB}^\pi$.

*Proof of Lemma 4.* We proof by verify the definition of stationary distribution. For any $(s_A', a_A')$, it holds

$$\sum_{s_A, a_A} \left( \sum_{s_B, a_B} \mu_{AB}^\pi(s_A, s_B, a_A, a_B) \right) P^\pi(s_A', a_A' \mid s_A, a_A)$$

$$= \sum_{s_A, a_A} \sum_{s_B, a_B} \mu_{AB}^\pi(s_A, s_B, a_A, a_B) \sum_{s_B', a_B'} \sum_{s_B'', a_B''} P^\pi\left(s_A', s_B', a_A', a_B' \mid s_A, s_B'', a_A, a_B''\right) \mu_{AB}^\pi(s_B'', a_B'' \mid s_A, a_A)$$

$$= \sum_{s_A, a_A} \sum_{s_B, a_B} \mu_{AB}^\pi(s_B, a_B \mid s_A, a_A) \sum_{s_B', a_B'} \sum_{s_B'', a_B''} P^\pi\left(s_A', s_B', a_A', a_B' \mid s_A, s_B'', a_A, a_B''\right) \mu_{AB}^\pi(s_A, s_B'', a_A, a_B'')$$

$$= \sum_{s_A, a_A} \sum_{s_B', a_B'} \sum_{s_B'', a_B''} P^\pi\left(s_A', s_B', a_A', a_B' \mid s_A, s_B'', a_A, a_B''\right) \mu_{AB}^\pi(s_A, s_B'', a_A, a_B'')$$

$$= \sum_{s_B', a_B'} \mu_{AB}^\pi(s_A', s_B', a_A', a_B') \,.$$

where the last equation follows from the definition of $\mu_{AB}^\pi$. Hence we conclude that $\sum_{s_B, a_B} \mu_{AB}^\pi(s_A, s_B, a_A, a_B)$ is a stationary distribution of $\boldsymbol{P}_A^\pi$. $\qquad\square$

We are then ready to prove Theorem 4. We first note that similar to ATV distance in Eq. (7), the optimal solution to $E_A(\boldsymbol{P}_{AB}^\pi)$ w.r.t $\mu_{AB}^\pi$-weighted ATV distance also only depends on $\boldsymbol{P}_A$. Thus, let $\boldsymbol{P}_A, \boldsymbol{P}_B$ be the optimal solutions to $E_A(\boldsymbol{P}_{AB}^\pi), E_B(\boldsymbol{P}_{AB}^\pi)$ respectively.

Let $\boldsymbol{x} \in \mathbb{R}^{|\mathcal{S}_A||\mathcal{A}_A|}$ with $\|\boldsymbol{x}\|_\infty = 1$. Following the same technique in the proof of Theorem 4, we have

$$\mu_A^{\pi^\top} |(\boldsymbol{P}_A^\pi - \boldsymbol{P}_A)\, \boldsymbol{x}|$$

$$= \sum_{s_A, a_A} \mu_A^\pi(s_A, a_A) \left| \sum_{s_A', a_A'} (\boldsymbol{P}_A^\pi - \boldsymbol{P}_A)_{(s_A', a_A' \mid s_A, a_A)} \, \boldsymbol{x}(s_A', a_A') \right|$$

$$= \sum_{s_A, a_A} \mu_A^\pi(s_A, a_A) \left| \sum_{s_A', a_A'} \boldsymbol{x}(s_A', a_A') \sum_{s_B', a_B'} \sum_{s_B, a_B} (\boldsymbol{P}_{AB}^\pi - \boldsymbol{P}_A \otimes \boldsymbol{P}_B)_{(\boldsymbol{s}', \boldsymbol{a}' \mid \boldsymbol{s})} \, \mu_{AB}^\pi(s_B, a_B \mid s_A, a_A) \right|$$

$$\leq \sum_{\boldsymbol{s}, \boldsymbol{a}} \left| \sum_{s_A', a_A'} \boldsymbol{x}(s_A', a_A') \sum_{s_B', a_B'} (\boldsymbol{P}_{AB}^\pi - \boldsymbol{P}_A \otimes \boldsymbol{P}_B)_{(\boldsymbol{s}', \boldsymbol{a}' \mid \boldsymbol{s}, \boldsymbol{a})} \right| \mu_{AB}^\pi(\boldsymbol{s}, \boldsymbol{a}) \leq 2E_A(\boldsymbol{P}_{AB}^\pi)$$

where the second last inequality follows from Lemma 4. We then conclude

$$\left\| \boldsymbol{P}_A^\pi - \boldsymbol{P}_A \right\|_{\mu, \infty} \leq 2E_A(\boldsymbol{P}_{AB}^\pi) \,,$$

and similar results hold for $\boldsymbol{P}_B^\pi$. We then apply the decomposition

$$(I - \gamma P_{AB}^\pi)^{-1} (r_A \otimes e) - \left( (I - \gamma P_A^\pi)^{-1} r_A \right) \otimes e$$

$$= \underbrace{(I - \gamma P_{AB}^\pi)^{-1} (r_A \otimes e) - (I - \gamma P_A \otimes P_B)^{-1} (r_A \otimes e)}_{(I)}$$

$$+ \underbrace{\left( (I - \gamma P_A)^{-1} r_A \right) \otimes e - \left( (I - \gamma P_A^\pi)^{-1} r_A \right) \otimes e}_{(II)}$$

For $(I)$, we have

$$\left\| (I - \gamma P_{AB}^\pi)^{-1} (r_A \otimes e) - (I - \gamma P_A \otimes P_B)^{-1} (r_A \otimes e) \right\|_{\mu_{AB}^\pi}$$

$$= \left\| (I - \gamma P_{AB}^\pi)^{-1} (\gamma P_{AB}^\pi - \gamma P_A \otimes P_B) (I - \gamma P_A \otimes P_B)^{-1} (r_A \otimes e) \right\|_{\mu_{AB}^\pi}$$

$$\overset{(i)}{\leq} \frac{1}{1 - \gamma} \left\| \left( (\gamma P_{AB}^\pi - \gamma P_A \otimes P_B) (I - \gamma P_A)^{-1} r_A \right) \otimes e \right\|_{\mu_{AB}^\pi}$$

$$\leq \frac{2\gamma E(\pi)}{1 - \gamma} \left\| (I - \gamma P_A)^{-1} r_A \right\|_\infty \leq \frac{2\gamma E(\pi) r_{\max}}{(1 - \gamma)^2},$$

where $(i)$ follows from the fact that for any $x$

$$\|Px\|_\mu = \mu^\top |Px| \leq \mu^\top P|x| = \mu^\top |x| = \|x\|_\mu.$$

For $(II)$ one can use Lemma 4 to verify

$$\left\| \left( (I - \gamma P_A)^{-1} r_A \right) \otimes e - \left( (I - \gamma P_A^\pi)^{-1} r_A \right) \otimes e \right\|_{\mu_{AB}^\pi}$$

$$= \left\| (I - \gamma P_A)^{-1} r_A - (I - \gamma P_A^\pi)^{-1} r_A \right\|_{\mu_A^\pi}$$

And similar results to $(I)$ holds. We then conclude the proof of Theorem 4.

## H    Results for multi-agent MDPs

In quantum physics, the concept of quantum entanglement of two-party system can be well extended to multi-party system. In this section, we demonstrate a similar extension of two-agent Markov entanglement to multi-agent settings. We begin with the model of multi-agent MDPs.

Consider an $N$-agent MDP $\mathcal{M}_{1:N}(\mathcal{S}, \mathcal{A}, P, r_{1:N}, \gamma)$ with joint state space $\mathcal{S} = \times_{i=1}^N \mathcal{S}_i$ and joint action space $\mathcal{A} = \times_{i=1}^N \mathcal{A}_i$. For simplicity, we assume $|\mathcal{S}_i| = |\mathcal{S}|$ and $|\mathcal{A}_i| = |\mathcal{A}|$ for each agent $i$. For agents at global state $s = (s_1, s_2, \ldots, s_N)$ with action $a = (a_1, a_2, \ldots, a_N)$ taken, the system will transit to $s' = (s_1', s_2', \ldots, s_N')$ according to transition kernel $s' \sim P(\cdot \mid s, a)$ and each agent $i \in [N]$ will receive its local reward $r_i(s_i, a_i)$. The global reward $r_{1:N}$ is defined as the summation of local rewards $r_{1:N}(s, a) := \sum_{i=1}^N r_i(s_i, a_i)$, or in vector form,

$$r_{1:N} \in \mathbb{R}^{|\mathcal{S}|^N |\mathcal{A}|^N} := \sum_{i=1}^N (e\otimes)^{i-1} r_i (\otimes e)^{N-i}.$$

We further assume the local rewards are bounded, i.e. for agent $i \in [N]$, $|r_i(s_i, a_i)| \leq r_{\max}^i$ for all $(s_i, a_i)$. Given any global policy $\pi \colon \mathcal{S} \to \Delta(\mathcal{A})$, we denote $P_{1:N}^\pi \in \mathbb{R}^{|\mathcal{S}|^N |\mathcal{A}|^N \times |\mathcal{S}|^N |\mathcal{A}|^N}$ as the transition matrix induced by $\pi$ where $P_{1:N}^\pi (s_{1:N}', a_{1:N}' \mid s_{1:N}, a_{1:N}) := P(s_{1:n}' \mid s_{1:N}, a_{1:N}) \pi(a_{1:N}' \mid s_{1:N}')$. Then the global Q-value is defined by Bellman Equation $Q_{1:N}^\pi = (I - \gamma P_{1:N}^\pi)^{-1} r_{1:N}$. The local Q-values follow the similar framework to Meta Algorithm 1 where each agent $i \in [N]$ fits $Q_i^\pi$ using its local observations. We then sum up local Q-values to approximate the global Q-value, i.e.

$$Q_{1:N}^\pi(s, a) \approx \sum_{i=1}^N Q_i^\pi(s_i, a_i).$$

To illustrate the extension, we first provide the definition of multi-party quantum entanglement here for reference.

**Definition 7** (Multi-party Quantum Entanglement). *Consider a multi-party quantum system composed of $N$ subsystems, indexed by $[N]$. The joint state $\rho_{1:N}$ is **separable** if there exists $K \in \mathbb{Z}^+$, probability distribution $\{x_i\}_{i \in [K]}$, and density matrices $\left\{\rho_{1:N}^{(j)}\right\}_{j \in [K]}$ such that*

$$\rho_{1:N} = \sum_{j=1}^K x_j \rho_1^{(j)} \otimes \rho_2^{(j)} \otimes \cdots \otimes \rho_N^{(j)}.$$

*If there exists no such decomposition, $\rho_{1:N}$ is called **entangled**.*

Analogically, we define the Multi-agent Markov Entanglement,

**Definition 8** (Multi-agent Markov Entanglement). *Consider a $N$-agent Markov system $\mathcal{M}_{1:N}$ and policy $\pi \colon \mathcal{S} \to \Delta(\mathcal{A})$, the agents are **separable** under policy $\pi$ if there exists $K \in \mathbb{Z}^+$, measure $\{x_j\}_{j \in [K]}$ satisfying $\sum_{j=1}^K x_j = 1$, and transition matrices $\left\{\boldsymbol{P}_{1:N}^{(j)}\right\}_{j \in [K]}$ such that*

$$\boldsymbol{P}_{1:N}^\pi = \sum_{j=1}^K x_j \boldsymbol{P}_1^{(j)} \otimes \boldsymbol{P}_2^{(j)} \otimes \cdots \otimes \boldsymbol{P}_N^{(j)}.$$

*If there exists no such decomposition, the agents are **entangled** under policy $\pi$.*

For clarity, we use superscript $s^i$ to denote the $i$-th element in state space and subscript $s_i$ to represent the state at $i$-th arm. Furthermore, we denote $\mathcal{S}^{-i} := \mathcal{S} \setminus s^i$ and $\boldsymbol{s} := s_{1:N} := \{s_1, s_2, \ldots, s_N\}$ is the profile of $N$-arms.

Given any global policy $\pi$, for any agent $i \in [N]$,

$$P_i^\pi(s_i', a_i' \mid s_i, a_i) = \sum_{s_{-i}', a_{-i}'} \sum_{s_{-i}, a_{-i}} P_{1:N}^\pi(s_{1:N}', a_{1:N}' \mid s_{1:N}, a_{1:N}) \rho_{1:N}^\pi(s_{-i}, a_{-i} \mid s_i, a_i).$$

**Definition 9** (Measure of Multi-agent Markov Entanglement). *Consider a $N$-agent Markov system $\mathcal{M}_{1:N}$ with joint state space $\mathcal{S} = \times_{i=1}^N \mathcal{S}_i$ and action space $\mathcal{A} = \times_{i=1}^N \mathcal{A}_i$. Given any policy $\pi \colon \mathcal{S} \to \Delta(\mathcal{A})$, the measure of Markov entanglement of $N$ agents is*

$$E(\boldsymbol{P}_{1:N}^\pi) = \min_{\boldsymbol{P} \in \mathcal{P}_{SEP}} d(\boldsymbol{P}_{1:N}^\pi, \boldsymbol{P}), \tag{13}$$

*where $d(\cdot, \cdot)$ is some distance measure.*

The following theorem generalizes the results of value-decomposition for two-agent Markov systems in Theorem 3 to multi-agent Markov systems.

**Theorem 7.** *Consider a $N$-agent MDP $\mathcal{M}_{1:N}$ with joint state space $\mathcal{S} = \times_{i=1}^N \mathcal{S}_i$ and action space $\mathcal{A} = \times_{i=1}^N \mathcal{A}_i$. Given any policy $\pi \colon \mathcal{S} \to \Delta(\mathcal{A})$ with the measure of Markov entanglement $E_i(\boldsymbol{P}_{1:N}^\pi)$ w.r.t ATV distance, it holds for any agent $i$,*

$$\|\boldsymbol{P}_i^\pi - \boldsymbol{P}_i\|_\infty \leq 2_i E(\boldsymbol{P}_{1:N}^\pi).$$

*where $\boldsymbol{P}_i$ is the optimal solution of Eq. (13). Furthermore, the decomposition error is entry-wise bounded by the measure of Markov entanglement,*

$$\left\| Q_{1:N}^\pi(\boldsymbol{s}, \boldsymbol{a}) - \sum_{i=1}^N Q_i^\pi(s_i, a_i) \right\|_\infty \leq \frac{4\gamma \left( \sum_{i=1}^N E_i(\boldsymbol{P}_{1:N}^\pi) r_{\max}^i \right)}{(1 - \gamma)^2}.$$

The proof mainly follows the following lemma, which generalizes the key technique used in Theorem 1.

**Lemma 5.** *For any agent $i$, it holds*

$$\left( \sum_{j=1}^K x_j \boldsymbol{P}_1^{(j)} \otimes \boldsymbol{P}_2^{(j)} \otimes \cdots \otimes \boldsymbol{P}_N^{(j)} \right) \cdot \left( (\boldsymbol{e}\otimes)^{i-1} \boldsymbol{r}_i (\otimes\boldsymbol{e})^{N-i} \right) = (\boldsymbol{e}\otimes)^{i-1} \left( \sum_{j=1}^K x_j \boldsymbol{P}_i^{(j)} \boldsymbol{r}_i \right) (\otimes\boldsymbol{e})^{N-i}. \tag{14}$$

The lemma follows from the property of tensor product. We can also extend Theorem 4 to multi-agent MDPs.

**Theorem 8.** *Consider a $N$-agent MDP $\mathcal{M}_{1:N}$ with joint state space $\mathcal{S} = \times_{i=1}^{N} \mathcal{S}_i$ and action space $\mathcal{A} = \times_{i=1}^{N} \mathcal{A}_i$. Given any policy $\pi \colon \mathcal{S} \to \Delta(\mathcal{A})$ with the measure of Markov entanglement $E_i(\boldsymbol{P}_{1:N}^{\pi})$ w.r.t the $\mu_{1:N}^{\pi}$-weighted agent-wise total variation distance, it holds for any agent $i$,*

$$\left\| \boldsymbol{P}_i^{\pi} - \boldsymbol{P}_i \right\|_{\mu_i^{\pi}, \infty} \le 2 E_i(\boldsymbol{P}_{1:N}^{\pi}) .$$

*where $\boldsymbol{P}_i$ is the optimal solution of Eq. (13) and $\mu_i^{\pi}$ is the stationary distribution of the projected transition $\boldsymbol{P}_i^{\pi}$. Furthermore, the $\mu_{1:N}^{\pi}$-weighted decomposition error is bounded by the measure of Markov entanglement,*

$$\left\| Q_{1:N}^{\pi}(\boldsymbol{s}, \boldsymbol{a}) - \sum_{i=1}^{N} Q_i^{\pi}(s_i, a_i) \right\|_{\mu_{1:N}^{\pi}} \le \frac{4\gamma \left( \sum_{i=1}^{N} E_i(\boldsymbol{P}_{1:N}^{\pi}) r_{\max}^{i} \right)}{(1 - \gamma)^2} .$$

# I  Proof of Theorem 6

We first provide an overview of the proof and introduce the technical assumptions.

To begin, we consider the system configuration $\boldsymbol{m} \in \Delta^{|\mathcal{S}|}$ where $\boldsymbol{m}_s = \frac{1}{N} \sharp \{\text{Agents in state s}\}$ is the proportion of agents in state $s$. When $N \to \infty$, the transition between configurations will become deterministic under index policy and $\boldsymbol{m}$ will approach its mean-field limit $\boldsymbol{m}^*$. Furthermore, in the mean-field, each agent's local transition will only depend its local state. As a result, the system will de-couple and become separable as $N \to \infty$.

To formalize this intuition, we introduce the following lemma that connects Markov entanglement measure with the mean-field analysis

**Lemma 6.** *The measure of Markov entanglement w.r.t $\mu_{1:N}^{\pi}$-weighted ATV distance is bounded by the deviation of mean-field configuration,*

$$E_i(\pi) \le |\mathcal{S}|^2 \cdot \mathbb{E}\left[ \|\boldsymbol{m} - \boldsymbol{m}^*\|_{\infty} \right] ,$$

*where the expectation is taking over the stationary distribution $\boldsymbol{m} \sim \mu_{1:N}^{\pi}$.*

We thus focus on the deviation from $\boldsymbol{m}$ to $\boldsymbol{m}^*$. We extend the concentration analysis from [20, 21] to derive a new stability bound for the RHS. Specifically, we finishing the proof via demonstrating the deviation decays at the rate $\mathcal{O}(1/\sqrt{N})$.

One caveat here is that we have to restrict chaotic behaviors in the mean-field limit. We thus introduce two technical assumptions.

We first define the transition of configuration under index policy $\pi$ as $\phi^{\pi} \colon \Delta^{|\mathcal{S}|} \to \Delta^{|\mathcal{S}|}$ such that

$$\phi^{\pi}(\boldsymbol{m}) = \mathbb{E}\left[ \boldsymbol{m}[t+1] \mid \boldsymbol{m}[t] = \boldsymbol{m}, \pi \right] .$$

For $t > 0$, we denote $\Phi_t := (\phi^{\pi})^t$ apply the transition mapping for $t$ rounds.

**Assumption A** (Uniform Global Attractor Property (UGAP)). *There exists a uniform global attractor $\boldsymbol{m}^*$ of $\phi^{\pi}(\cdot)$, i.e. for all $\varepsilon > 0$, there exists $T(\varepsilon)$ such that for all $t \ge T(\varepsilon)$ and all $\boldsymbol{m} \in \Delta^{|\mathcal{S}|}$, one has $\|\Phi_t(\boldsymbol{m}) - \boldsymbol{m}^*\|_{\infty} < \varepsilon$.*

The UGAP assumption ensures the uniqueness of $\boldsymbol{m}^*$ and guarantees fast convergence from any initial $\boldsymbol{m}$ to $\boldsymbol{m}^*$.

**Assumption B** (Non-degenerate RMAB). *There exists state $s \in \mathcal{S}$ such that $0 < \pi^*(s, 0) < 1$, where $\pi^*$ is the policy under $\boldsymbol{m}^*$.*

The non-degenerate assumption further restricts cyclic behavior in the mean-field limit.

Non-degenerate and UGAP are two standard technical assumptions for the index policy, which restrict chaotic behavior in asymptotic regime and will be further introduced in subsequent sections. We note here these two assumptions are also used in almost all theoretical work on index policies [47, 41, 20, 21].

*Proof of Theorem 6.* In the subsequent proof, we let $\nu_1 > \nu_2 > \nu_3 > \cdots > \nu_{|S|}$. This does not lose generality in that we can always exchange state index. The proof consists of several steps

**Step 1: Find $m^*$**  Recall the transition mapping for configurations $\phi^\pi : \Delta^{|\mathcal{S}|} \to \Delta^{|\mathcal{S}|}$,
$$\phi^\pi(m) = \mathbb{E}\left[m[t+1] \mid m[t] = m, \pi\right].$$
Notice that the definition of $\phi^\pi$ does not depend on $N$. We adapt from Lemma B.1 in [20] defined specially for Whittle Index,

**Lemma 7** (Piecewise Affine). *Given any index policy $\pi$, $\phi^\pi$ is a piecewise affine continuous function with $|\mathcal{S}|$ affine pieces.*

When the context is clear, we abbreviate $\phi^\pi$ as $\phi$. For any $m \in \Delta^{|\mathcal{S}|}$, define $s(m) \in [|\mathcal{S}|]$ be the state such that $\sum_{i=1}^{s(m)-1} m_i \leq \alpha < \sum_{i=1}^{s(m)} m_i$. Lemma 7 characterizes for any $m \in \mathcal{Z}_i :=$ $\left\{m \in \Delta^{|\mathcal{S}|} \mid s(m) = i\right\}$, there exists $K_{s(m)}, b_{s(m)}$ such that
$$\phi(m) = K_{s(m)}m + b_{s(m)}.$$
By Brouwer fixed point theorem, there exists a fixed point $m^*$ such that $\phi(m^*) = m^*$. The UGAP condition guarantees the uniqueness of $m^*$. Our choice of $\pi^*$ is the corresponding policy under $m^*$.

**Step 2: Connecting policy entanglement with the deviation of stationary distribution**  Combine Proposition 9 with the RMAB model, we have

**Lemma 8.** *The measure of Markov entanglement w.r.t $\mu_{1:N}^\pi$-weighted ATV distance is bounded by the deviation of mean-field configuration,*
$$E_i(\pi) \leq |\mathcal{S}|^2 \cdot \mathbb{E}\left[\|m - m^*\|_\infty\right],$$
*where the expectation is taking over the stationary distribution $m \sim \mu_{1:N}^\pi$.*

*Proof.* Given the homogeneity of agents, we first demonstrate for any two agent $i, j$, it holds
$$\sum_{s_{1:N}} \mu^\pi(s_{1:N}) \left|\pi(a_i = a \mid s_{1:N}) - \pi^*(a_i = a \mid s_i)\right| = \sum_{s_{1:N}} \mu^\pi(s_{1:N}) \left|\pi(a_j = a \mid s_{1:N}) - \pi^*(a_j = a \mid s_i)\right|.$$
To see this, we first notice by the definition of index policy
$$\left|\pi(a_i = a \mid s_i = s, m) - \pi^*(a \mid s)\right| = \left|\pi(a_j = a \mid s_j = s, m) - \pi^*(a \mid s)\right|.$$
It then suffices to prove $\sum_{s_i = s, s_{1:N} = m} \mu(s_{1:N}) = \sum_{s_j = s, s_{1:N} = m} \mu(s_{1:N})$. If $\sum_{s_i = s, s_{1:N} = m} \mu(s_{1:N}) \leq \sum_{s_j = s, s_{1:N} = m} \mu(s_{1:N})$, we can exchange the agent index of $i$ and $j$. This will result in the same stationary distribution and $\sum_{s_i = s, s_{1:N} = m} \mu(s_{1:N}) \geq \sum_{s_j = s, s_{1:N} = m} \mu(s_{1:N})$ and thus the equation. We then rewrite the bound in Proposition 9,
$$\begin{aligned}
E(\pi) &\leq \frac{1}{2} \sup_i \sum_{s_{1:N}} \mu^\pi(s_{1:N}) \sum_{a_i} \left|\pi(a_i \mid s_{1:N}) - \pi^*(a_i \mid s_i)\right| \\
&= \sup_i \sum_{s_{1:N}} \mu^\pi(s_{1:N}) \left|\pi(a_i = 1 \mid s_{1:N}) - \pi^*(a_i = 1 \mid s_i)\right| \\
&= \frac{1}{N} \sum_{s_{1:N}} \mu^\pi(s_{1:N}) \sum_{i=1}^N \left|\pi(a_i = 1 \mid s_{1:N}) - \pi^*(a_i = 1 \mid s_i)\right| \\
&= \sum_m \mu^\pi(m) \sum_{s \in \mathcal{S}} m_s \left|\pi(a = 1 \mid s, m) - \pi^*(a = 1 \mid s)\right|
\end{aligned}$$
For any configuration $m$ and state $s$, we have
$$\begin{aligned}
&m_s \left|\pi(a = 1 \mid s, m) - \pi^*(a = 1 \mid s)\right| \\
=& m_s \left|\frac{\pi^*(a = 1 \mid s)m_s^* N + k_s}{m_s^* N + \ell_s} - \pi^*(a = 1 \mid s)\right| \\
=& \frac{m_s^* N + \ell_s}{N} \left|\frac{k_s - \ell_s \pi^*(a = 1 \mid s)}{m_s^* N + \ell_s}\right| \\
\leq& |\mathcal{S}| \|m - m^*\|_\infty,
\end{aligned}$$
where $|k_s| \leq (|\mathcal{S}| - 1)\|m - m^*\|_\infty N$ representing the additional fraction of state $s$ to be activated due to the deviation from $m^*$ and $|\ell_s| \leq \|m - m^*\|_\infty N$ representing the deviation of $m_s$ from $m_s^*$. The results then hold by taking summation over $s$ and expectation over $m$.

$\square$

**Step 3: Concentrations and local stability**  To bound $\mathbb{E}\left[\|\boldsymbol{m} - \boldsymbol{m}^*\|_\infty\right]$, we start with several technical lemmas from previous RMAB literature. We use the same notation $\Phi_t = \phi(\Phi_{t-1})$.

**Lemma 9** (One-step Concentration, Lemma 1 in [21]). *Let $\epsilon[1] = \boldsymbol{m}[1] - \phi(\boldsymbol{m}[0])$, it holds*

$$\mathbb{E}\left[\|\epsilon[1]\|_1 \mid \boldsymbol{m}[0]\right] \leq \sqrt{\frac{|\mathcal{S}|}{N}} .$$

**Lemma 10** (Multi-step Concentration, Lemma C.4 in [20]). *There exists a positive constant $K$ such that for all $t \in \mathbb{N}$ and $\delta > 0$,*

$$\Pr\left[\|\boldsymbol{m}[t] - \Phi_t(\boldsymbol{m})\|_\infty \geq (1 + K + K^2 + \cdots + K^t)\delta \mid \boldsymbol{m}[0] = \boldsymbol{m}\right] \leq t|\mathcal{S}|e^{-2N\delta^2}$$

**Lemma 11** (Local Stability, Lemma C.5 in [20]). *Under non-degenerate and UGAP:*

  (i) $\boldsymbol{K}_{s(\boldsymbol{m}^*)}$ *is a stable matrix, i.e. its spectral radius is strictly less than* 1.

  (ii) *For any $\epsilon$, there exists $T(\epsilon) > 0$ such that for all $\boldsymbol{m} \in \Delta^{|\mathcal{S}|}$, $\left\|\Phi_{T(\epsilon)}(\boldsymbol{m}) - \boldsymbol{m}^*\right\|_\infty < \epsilon$.*

The first result implies there exists some matrix norm $\|\cdot\|_\beta$ such that $\left\|\boldsymbol{K}_{s(\boldsymbol{m}^*)}\right\|_\beta < 1$. By the equivalence of norms, there exists constant $C^1_\beta, C^2_\beta > 0$ such that for all $\boldsymbol{x} \in \mathbb{R}^{|\mathcal{S}|}$

$$C^1_\beta\|\boldsymbol{x}\|_\beta \leq \|\boldsymbol{x}\|_\infty \leq C^2_\beta\|\boldsymbol{x}\|_\beta .$$

Combine the second result of Lemma 11 and non-degenerate condition, we can construct a neighborhood $\mathcal{N}$ of $\boldsymbol{m}^*$ such that $\mathcal{N} = \mathcal{B}(\boldsymbol{m}^*, \epsilon) \cap \Delta^{|\mathcal{S}|} \in \mathcal{Z}_{s(\boldsymbol{m}^*)}$ where $\epsilon > 0$ and $\mathcal{B}(\boldsymbol{m}^*, \epsilon) = \{\boldsymbol{m} \mid \|\boldsymbol{m} - \boldsymbol{m}^*\|_\infty < \epsilon\}$ is an open ball. We next show that $\boldsymbol{m}[0]$ under stationary distribution will concentrate in $\mathcal{N}$ with high probability. Let $\tilde{T} = T(\epsilon/2)$ such that for all $\boldsymbol{m} \in \Delta^{|\mathcal{S}|}$, $\|\Phi_{\tilde{T}}(\boldsymbol{m}) - \boldsymbol{m}^*\|_\infty < \epsilon/2$. It holds

$$
\begin{aligned}
\Pr\left[\boldsymbol{m}[0] \neq \mathcal{N}\right] = \Pr\left[\|\boldsymbol{m}[0] - \boldsymbol{m}^*\|_\infty \geq \epsilon\right] \\
\overset{(i)}{=} \Pr\left[\left\|\boldsymbol{m}[\tilde{T}] - \boldsymbol{m}^*\right\|_\infty \geq \epsilon \mid \boldsymbol{m}[0] = \boldsymbol{m}\right] \\
\leq \Pr\left[\left\|\boldsymbol{m}[\tilde{T}] - \Phi_{\tilde{T}}(\boldsymbol{m})\right\|_\infty \geq \frac{\epsilon}{2} \mid \boldsymbol{m}[0] = \boldsymbol{m}\right] + \Pr\left[\|\Phi_{\tilde{T}}(\boldsymbol{m}) - \boldsymbol{m}^*\|_\infty \geq \frac{\epsilon}{2}\right] \\
= \Pr\left[\left\|\boldsymbol{m}[\tilde{T}] - \Phi_{\tilde{T}}(\boldsymbol{m})\right\|_\infty \geq \frac{\epsilon}{2} \mid \boldsymbol{m}[0] = \boldsymbol{m}\right] \leq \tilde{T}|\mathcal{S}|e^{-2uN}
\end{aligned}
$$

where $(i)$ follows from the stationarity $\boldsymbol{m}[\tilde{T}]$ and $\boldsymbol{m}[0]$ are *i.i.d* and the constant $u = \left(\frac{\epsilon}{2(1+K+K^2+\cdots+K^{\tilde{T}})}\right)^2$ does not depend on $N$.

**Step 4: Put it together**  Finally, we are ready to bound $\mathbb{E}\left[\|\boldsymbol{m} - \boldsymbol{m}^*\|_\infty\right]$. Notice for all $\boldsymbol{m}[0] \in \mathcal{N}$, we have

$$
\begin{aligned}
\boldsymbol{m}[1] - \boldsymbol{m}^* &= \phi(\boldsymbol{m}[0]) + \epsilon[1] - \boldsymbol{m}^* \\
&= \boldsymbol{K}_{s(\boldsymbol{m}^*)}\left(\boldsymbol{m}[0] - \boldsymbol{m}^*\right) + \epsilon[1] .
\end{aligned}
$$

Taking $\|\cdot\|_\beta$ on both side,

$$
\begin{aligned}
\|\boldsymbol{m}[1] - \boldsymbol{m}^*\|_\beta &\leq \left\|\boldsymbol{K}_{s(\boldsymbol{m}^*)}\left(\boldsymbol{m}[0] - \boldsymbol{m}^*\right)\right\|_\beta + \|\epsilon[1]\|_\beta \\
&\leq \left\|\boldsymbol{K}_{s(\boldsymbol{m}^*)}\right\|_\beta \|\boldsymbol{m}[0] - \boldsymbol{m}^*\|_\beta + \|\epsilon[1]\|_\beta .
\end{aligned}
$$

Taking expectation on both side,

$$
\begin{aligned}
&\mathbb{E}\left[\|\boldsymbol{m}[1] - \boldsymbol{m}^*\|_\beta\right] \\
=&\mathbb{E}\left[\|\phi(\boldsymbol{m}[0]) - \boldsymbol{m}^*\|_\beta \cdot \mathbf{1}\left\{\boldsymbol{m}[0] \in \mathcal{N}\right\}\right] + \mathbb{E}\left[\|\phi(\boldsymbol{m}[0]) - \boldsymbol{m}^*\|_\beta \cdot \mathbf{1}\left\{\boldsymbol{m}[0] \notin \mathcal{N}\right\}\right] + \mathbb{E}\left[\|\epsilon[1]\|_\beta\right] \\
\leq& \left\|\boldsymbol{K}_{s(\boldsymbol{m}^*)}\right\|_\beta \mathbb{E}\left[\|\boldsymbol{m}[0] - \boldsymbol{m}^*\|_\beta \cdot \mathbf{1}\left\{\boldsymbol{m}[0] \in \mathcal{N}\right\}\right] + \Pr\left[\boldsymbol{m}[0] \notin \mathcal{N}\right]\sup_{\boldsymbol{m}[0]}\|\phi(\boldsymbol{m}[0]) - \boldsymbol{m}^*\|_\beta + \mathbb{E}\left[\|\epsilon[1]\|_\beta\right] \\
\leq& \left\|\boldsymbol{K}_{s(\boldsymbol{m}^*)}\right\|_\beta \mathbb{E}\left[\|\boldsymbol{m}[0] - \boldsymbol{m}^*\|_\beta\right] + \Pr\left[\boldsymbol{m}[0] \notin \mathcal{N}\right]\sup_{\boldsymbol{m}[0]}\|\phi(\boldsymbol{m}[0]) - \boldsymbol{m}^*\|_\beta + \mathbb{E}\left[\|\epsilon[1]\|_\beta\right] .
\end{aligned}
$$

By stationarity, one have $\mathbb{E}\left[\|\boldsymbol{m}[1] - \boldsymbol{m}^*\|_\beta\right] = \mathbb{E}\left[\|\boldsymbol{m}[0] - \boldsymbol{m}^*\|_\beta\right]$. This refines the above inequality,

$$
\begin{aligned}
\mathbb{E}\left[\|\boldsymbol{m}[0] - \boldsymbol{m}^*\|_\infty\right] &\leq \frac{C_\beta^2}{1 - \left\|\boldsymbol{K}_{s(\boldsymbol{m}^*)}\right\|_\beta}\left(\sup_{\boldsymbol{m}[0]} \Pr\left[\boldsymbol{m}[0] \notin \mathcal{N}\right]\|\phi(\boldsymbol{m}[0]) - \boldsymbol{m}^*\|_\beta + \mathbb{E}\left[\|\epsilon[1]\|_\beta\right]\right) \\
&\leq \frac{C_\beta^2}{C_\beta^1(1 - \left\|\boldsymbol{K}_{s(\boldsymbol{m}^*)}\right\|_\beta)}\left(\Pr\left[\boldsymbol{m}[0] \notin \mathcal{N}\right] + \mathbb{E}\left[\|\epsilon[1]\|_\infty\right]\right) \\
&\leq \frac{C_\beta^2}{C_\beta^1(1 - \left\|\boldsymbol{K}_{s(\boldsymbol{m}^*)}\right\|_\beta)}\left(\tilde{T}|\mathcal{S}|e^{-2uN} + \frac{\sqrt{|\mathcal{S}|}}{\sqrt{N}}\right).
\end{aligned}
$$

We combine Lemma 8 and conclude the proof of Theorem 6.

# J  Extensions of Markov entanglement

## J.1  (Weakly-)coupled MDPs

Weakly-coupled MDPs (WCMDP) are a rich class of multi-agent model that capture many real-world applications such as supply chain management, queuing network and resource allocations [2, 12, 36]. Compared to general multi-agent MDP, WCMDP further ensures each agent follow its local transition while the agents' actions are coupled with each other. Formally,

**Definition 10** (Weakly-coupled MDPs). *An $N$-agent MDP $\mathcal{M}_{1:N}(\mathcal{S}, \mathcal{A}, \boldsymbol{P}, \boldsymbol{r}_{1:N}, \gamma)$ is a weakly-coupled MDP if*

- *Each agent has local transition kernel $\boldsymbol{P}_i$ such that $\forall \boldsymbol{s}, \boldsymbol{a}, \boldsymbol{s}', P(\boldsymbol{s}' \mid \boldsymbol{s}, \boldsymbol{a}) = \prod_{i=1}^N P_i(s_i' \mid s_i, a_i)$.*

- *At global state $\boldsymbol{s}$, agents' joint actions $\boldsymbol{a}$ are subject to $m$ coupling constraints $\sum_{i=1}^N \boldsymbol{d}(s_i, a_i) \leq \boldsymbol{b} \in \mathbb{R}^m$.*

We then demonstrate that this weakly-coupled structure can further refine the analysis of Markov entanglement measure.

**Proposition 9.** *Consider a $N$-agent weakly-coupled MDP $\mathcal{M}_{1:N}(\mathcal{S}, \mathcal{A}, \boldsymbol{P}, \boldsymbol{r}_{1:N}, \gamma)$. Given any policy $\pi\colon \mathcal{S} \to \Delta(\mathcal{A})$ with measure of Markov entanglement $E_i(\boldsymbol{P}_{1:N}^\pi)$ w.r.t the $\mu_{1:N}^\pi$-weighted agent-wise total variation distance, it holds for $i \in [N]$,*

$$
E_i(\boldsymbol{P}_{1:N}^\pi) \leq \min_{\pi'} \frac{1}{2} \sum_{\boldsymbol{s}} \mu_{1:N}^\pi(\boldsymbol{s}) \sum_{a_i} |\pi(a_i \mid \boldsymbol{s}) - \pi'(a_i \mid s_i)| \ ,
$$

*where $\pi' : \mathcal{S}_i \to \mathcal{A}_i$ is any local policy for agent $i$.*

*Proof of Proposition 9.* We demonstrate the proof for two-agent WCMDP and the generalization to multi-agent WCMDP is straightforward. Consider $\boldsymbol{P}_A^{\pi'}$ be the transition of agent $A$ under local policy

$\pi'$. We focus on agent $A$

$$E_A(\boldsymbol{P}_{AB}^\pi)$$

$$\leq \frac{1}{2} \sum_{\boldsymbol{s},\boldsymbol{a}} \mu_{AB}^\pi(\boldsymbol{s},\boldsymbol{a}) \sum_{s_A',a_A'} \left| P_{AB}^\pi(s_A',a_A' \mid \boldsymbol{s},\boldsymbol{a}) - P_A^{\pi'}(s_A',a_A' \mid s_A,a_A) \right|$$

$$= \frac{1}{2} \sum_{\boldsymbol{s},\boldsymbol{a}} \mu_{AB}^\pi(\boldsymbol{s},\boldsymbol{a}) \sum_{s_A',a_A'} \left| \sum_{s_B'} P_{AB}^\pi(\boldsymbol{s}',a_A \mid \boldsymbol{s},\boldsymbol{a}) - P_A^{\pi'}(s_A' \mid s_A,a_A)\pi'(a_A' \mid s_A') \right|$$

$$\overset{(i)}{=} \frac{1}{2} \sum_{\boldsymbol{s},\boldsymbol{a}} \mu_{AB}^\pi(\boldsymbol{s},\boldsymbol{a}) \sum_{s_A',a_A'} \left| \sum_{s_B'} P_{AB}^\pi(\boldsymbol{s}',a_A \mid \boldsymbol{s},\boldsymbol{a}) - \sum_{s_B'} P(\boldsymbol{s}' \mid \boldsymbol{s},\boldsymbol{a})\pi'(a_A' \mid s_A') \right|$$

$$= \frac{1}{2} \sum_{\boldsymbol{s},\boldsymbol{a}} \mu_{AB}^\pi(\boldsymbol{s},\boldsymbol{a}) \sum_{s_A',a_A'} \left| \sum_{s_B'} P(\boldsymbol{s}' \mid \boldsymbol{s},\boldsymbol{a}) \left( \pi(a_A' \mid \boldsymbol{s}') - \pi'(a_A' \mid s_A') \right) \right|$$

$$\leq \frac{1}{2} \sum_{\boldsymbol{s},\boldsymbol{a}} \mu_{AB}^\pi(\boldsymbol{s},\boldsymbol{a}) \sum_{\boldsymbol{s}'} P(\boldsymbol{s}' \mid \boldsymbol{s},\boldsymbol{a}) \sum_{a_A'} |\pi(a_A' \mid \boldsymbol{s}') - \pi'(a_A' \mid s_A')|$$

$$\overset{(ii)}{=} \frac{1}{2} \sum_{\boldsymbol{s}'} \mu_{AB}^\pi(\boldsymbol{s}') \sum_{a_A'} |\pi(a_A' \mid \boldsymbol{s}') - \pi'(a_A' \mid s_A')| \,.$$

where $(i)$ follows from the transition structure of weakly coupled MDP $P(\boldsymbol{s}' \mid \boldsymbol{s},\boldsymbol{a}) = P(s_A' \mid s_A,a_A) \cdot P(s_B' \mid s_B,a_B)$; and $(ii)$ comes from the fact that $P^\pi(\boldsymbol{s}' \mid \boldsymbol{s}) = \sum_{\boldsymbol{a}} \pi(\boldsymbol{a} \mid \boldsymbol{s}) P(\boldsymbol{s}' \mid \boldsymbol{s},\boldsymbol{a})$ and $\sum_{\boldsymbol{s}} \mu^\pi(\boldsymbol{s}) P^\pi(\boldsymbol{s}' \mid \boldsymbol{s}) = \mu^\pi(\boldsymbol{s}')$. $\qquad\square$

Proposition 9 establishes an upper bound for Markov entanglement in WCMDP. Intuitively, this bound characterizes *how agent $i$ can be viewed as making independent decisions*. It takes advantage of the weakly-coupled structure and shaves off the transition in Markov entanglement measure.

## J.2 Coupled MDPs with exogenous information

In many practical scenarios, the agents' transitions and actions are coupled by a shared exogenous signal. For example, in ride-hailing platforms, the specific dispatch is related to the exogenous order at the current moment [33, 24, 4]; in warehouse routing, the scheduling of robots is also related to the exogenous task revealed so far [15].

We will then enrich our framework by incorporating these exogenous information. At each timestep $t$, there will an exogenous information $z_t$ revealed to the decision maker. $z_t$ is assumed to evolve following a Markov chain independent of the action and transition of agents. We assume $z_t \in \mathcal{Z}$ and $\mathcal{Z}$ is finite.

Given the current state $\boldsymbol{s}$ and exogenous information $z$, the policy is given by $\pi : \mathcal{S} \times \mathcal{Z} \to \Delta(\tilde{\mathcal{A}})$, where $\tilde{\mathcal{A}}$ refers to the set of feasible actions. We then have the global transition depending on exogenous information $z$,

$$P_{ABz}^\pi(\boldsymbol{s}',\boldsymbol{a}',z' \mid \boldsymbol{s},\boldsymbol{a},z) = P(\boldsymbol{s}' \mid \boldsymbol{s},\boldsymbol{a},z) \cdot \pi(\boldsymbol{a}' \mid \boldsymbol{s}',z') \cdot P(z' \mid z) \,.$$

and global Q-value $Q_{ABz}^\pi \in \mathbb{R}^{|\mathcal{S}|^N |\mathcal{A}|^N |\mathcal{Z}|}$,

$$Q_{AB}^\pi(\boldsymbol{s},\boldsymbol{a},z) = \mathbb{E}\left[ \sum_{t=0}^\infty \sum_{i=1}^N r(s_{i,t},a_{i,t},z_t) \mid \boldsymbol{s}_0 = \boldsymbol{s}, \boldsymbol{a}_0 = \boldsymbol{a}, z_0 = z \right] \,.$$

We assume the system is unichain and the stationary distribution is $\mu_{ABz}^\pi$. Then we can derive the local transition under new algorithm by

$$P_{Az}(s_A',a_A',z' \mid s_A,a_A,z) = \sum_{s_B,a_B} \mu_{ABz}^\pi(s_B,a_B \mid s_A,a_A,z) \sum_{s_B',a_B'} P_{ABz}^\pi(\boldsymbol{s}',\boldsymbol{a}',z' \mid \boldsymbol{s},\boldsymbol{a},z) \,,$$

Given the local transition, we have the local value $\boldsymbol{Q}_{Az}^\pi = (\boldsymbol{I} - \gamma \boldsymbol{P}_{Az})^{-1}(\boldsymbol{r}_{Az})$ via Bellman Equation.

Combined with exogenous information, we consider the following value decomposition

$$Q_{AB}^{\pi}(\boldsymbol{s}, \boldsymbol{a}, z) = Q_A^{\pi}(s_A, a_A, z) + Q_B^{\pi}(s_B, a_B, z).$$

We start by introducing agent-wise Markov entanglement defined for each agent

$$\boldsymbol{P}_{ABz}^{\pi} = \sum_{j=1}^{K} x_j \boldsymbol{P}_{Az}^{(j)} \otimes \boldsymbol{P}_B^{(j)}. \tag{15}$$

**Proposition 10.** *If the system is agent-wise separable for all agents, then*

$$\boldsymbol{Q}_{ABz}^{\pi} = \boldsymbol{Q}_{Az}^{\pi} \otimes \boldsymbol{e}_{|\mathcal{S}||\mathcal{A}|} + \boldsymbol{e}_{|\mathcal{S}||\mathcal{A}|} \otimes \boldsymbol{Q}_{Bz}^{\pi}.$$

*Proof.* The proof is basically the same as Theorem 1. One can first quickly show that $\boldsymbol{P}_{Az} = \sum_{j=1}^{K} x_j \boldsymbol{P}_{Az}^{(j)}$. And then it holds

$$\left( \sum_{j=1}^{K} x_j \boldsymbol{P}_{Az}^{(j)} \otimes \boldsymbol{P}_B^{(j)} \right)^t \left( \boldsymbol{r}_A \otimes \boldsymbol{e}_{|z|} \otimes \boldsymbol{e}_{|\mathcal{S}||\mathcal{A}|} \right)$$

$$= \left( \sum_{j=1}^{K} x_j \boldsymbol{P}_{Az}^{(j)} \otimes \boldsymbol{P}_B^{(j)} \right)^{t-1} \left( \sum_{j=1}^{K} x_j \left( \boldsymbol{P}_{Az}^{(j)} (\boldsymbol{r}_A \otimes \boldsymbol{e}_{|z|}) \right) \otimes \left( \boldsymbol{P}_B^{(j)} \boldsymbol{e} \right) \right)$$

$$= \left( \sum_{j=1}^{K} x_j \boldsymbol{P}_{Az}^{(j)} \otimes \boldsymbol{P}_B^{(j)} \right)^{t-1} \left( \sum_{j=1}^{K} x_j \boldsymbol{P}_{Az}^{(j)} (\boldsymbol{r}_A \otimes \boldsymbol{e}_{|z|}) \right) \otimes \boldsymbol{e}$$

$$= \ldots = \left( \left( \sum_{j=1}^{K} x_j \boldsymbol{P}_{Az}^{(j)} \right)^t (\boldsymbol{r}_A \otimes \boldsymbol{e}_{|z|}) \right) \otimes \boldsymbol{e}.$$

$\square$

We then provide the measure of Markov entanglement with exogenous information w.r.t agent-wise total variation distance.

$$E_A(\boldsymbol{P}_{AB}^{\pi}, \mathcal{Z}) := \min \frac{1}{2} \left\| \boldsymbol{P}_{ABz}^{\pi} - \sum_{j=1}^{K} x_j \boldsymbol{P}_{Az}^{(j)} \otimes \boldsymbol{P}_B^{(j)} \right\|_{\mathrm{ATV}_1}$$

$$= \min_{\boldsymbol{P}_{Az}} \max_{\boldsymbol{s}, \boldsymbol{a}, z} \frac{1}{2} \sum_{s_A', a_A', z'} |P_{ABz}^{\pi}(s_A', a_A', z' \mid \boldsymbol{s}, \boldsymbol{a}, z) - P_{Az}(s_A', a_A', z' \mid s_A, a_A, z)|.$$

$$(16)$$

Similar to Theorem 3, we can connect this measure of Markov entanglement with the value decomposition error.

**Theorem 11.** *Consider a $N$-agent Markov system $\mathcal{M}_{1:N}$. Given any policy $\pi \colon \mathcal{S} \to \Delta(\mathcal{A})$ with the measure of Markov entanglement $E_i(\boldsymbol{P}_{1:N}^{\pi}, \mathcal{Z})$ w.r.t the agent-wise total variation distance, it holds for any agent $i$,*

$$\left\| \boldsymbol{P}_{iz}^{\pi} - \sum_{j=1}^{K} x_j \boldsymbol{P}_{iz}^{(j)} \right\|_{\infty} \leq 2E_i(\boldsymbol{P}_{1:N}^{\pi}, \mathcal{Z}).$$

*Furthermore, the decomposition error is entry-wise bounded by the measure of Markov entanglement,*

$$\left\| Q_{1:N}^{\pi}(\boldsymbol{s}, \boldsymbol{a}, z) - \sum_{i=1}^{N} Q_{iz}^{\pi}(s_i, a_i, z) \right\|_{\infty} \leq \frac{4\gamma \left( \sum_{i=1}^{N} E_i(\boldsymbol{P}_{1:N}^{\pi}, \mathcal{Z}) r_{\max}^i \right)}{(1-\gamma)^2}.$$

In practice, exogenous information is often discussed in the context of (weakly-)coupled MDPs, where each agent independent evolves by $P_i(s_{i+1} \mid s_i, a_i, z)$. Interestingly, we can derive a similar result to Proposition 9 that shaves off the transition in entanglement analysis.

**Proposition 12.** *Consider a $N$-agent Weakly Coupled Markov system $\mathcal{M}_{1:N}$. Given any policy $\pi: \mathcal{S} \to \Delta(\mathcal{A})$ and its measure of Markov entanglement $E_i(\boldsymbol{P}_{1:N}^\pi, \mathcal{Z})$ w.r.t the $\mu_{1:N}^\pi$-weighted agent-wise total variation distance, it holds*

$$E_i(\boldsymbol{P}_{1:N}^\pi, \mathcal{Z}) \leq \frac{1}{2} \sum_{s_{1:N}, z} \mu^\pi(s_{1:N}, z) \sum_{a_i} |\pi(a_i \mid s_{1:N}, z) - \pi'(a_i \mid s_i, z)| \,,$$

*for any policies $\pi'$.*

*Proof.* We provide the proof for two-agent MDP, which can be easily generalized to $N$-agent case.

$$
\begin{aligned}
&E_A(\boldsymbol{P}_{AB}^\pi, \mathcal{Z}) \\
&\leq \frac{1}{2} \sum_{\boldsymbol{s}, \boldsymbol{a}, z} \mu(\boldsymbol{s}, \boldsymbol{a}, z) \sum_{s'_A, a'_A, z'} |P_{ABz}^\pi(s'_A, a'_A, z' \mid \boldsymbol{s}, \boldsymbol{a}, z) - P_{Az}(s'_A, a'_A, z' \mid s_A, a_A, z)| \\
&= \frac{1}{2} \sum_{\boldsymbol{s}, \boldsymbol{a}, z} \mu(\boldsymbol{s}, \boldsymbol{a}, z) \sum_{s'_A, a'_A, z'} \left| \sum_{s'_B} P_{ABz}^\pi(\boldsymbol{s}', a_A, z' \mid \boldsymbol{s}, \boldsymbol{a}, z) - P_{Az}(s'_A, z' \mid s_A, a_A, z)\pi'(a'_A \mid s'_A, z') \right| \\
&= \frac{1}{2} \sum_{\boldsymbol{s}, \boldsymbol{a}, z} \mu(\boldsymbol{s}, \boldsymbol{a}, z) \sum_{s'_A, a'_A, z'} \left| \sum_{s'_B} P_{ABz}^\pi(\boldsymbol{s}', a_A, z' \mid \boldsymbol{s}, \boldsymbol{a}, z) - \sum_{s'_B} P(\boldsymbol{s}', z' \mid \boldsymbol{s}, \boldsymbol{a}, z)\pi'(a'_A \mid s'_A, z') \right| \\
&= \frac{1}{2} \sum_{\boldsymbol{s}, \boldsymbol{a}, z} \mu(\boldsymbol{s}, \boldsymbol{a}, z) \sum_{s'_A, a'_A, z'} \left| \sum_{s'_B} P(\boldsymbol{s}', z' \mid \boldsymbol{s}, \boldsymbol{a}, z) \left( \pi(a'_A \mid \boldsymbol{s}', z') - \pi'(a'_A \mid s'_A, z') \right) \right| \\
&\leq \frac{1}{2} \sum_{\boldsymbol{s}, \boldsymbol{a}, z} \mu(\boldsymbol{s}, \boldsymbol{a}, z) \sum_{\boldsymbol{s}', z'} P(\boldsymbol{s}', z' \mid \boldsymbol{s}, \boldsymbol{a}, z) \sum_{a'_A} |\pi(a'_A \mid \boldsymbol{s}', z') - \pi'(a'_A \mid s'_A, z')| \\
&= \frac{1}{2} \sum_{\boldsymbol{s}', z'} \mu(\boldsymbol{s}', z') \sum_{a'_A} |\pi(a'_A \mid \boldsymbol{s}', z') - \pi'(a'_A \mid s'_A, z')| \,.
\end{aligned}
$$

$\square$

## J.3 Factored MDPs

Another common class of multi-agent MDPs is Factored MDPs (FMDPs, [22, 23, 32]), which explicitly model the structured dependencies in state transitions. For instance, in a server cluster, the state transition of each server depends only on its neighboring servers. Formally, we define

**Definition 11** (Factored MDPs). *An $N$-agent MDP $\mathcal{M}_{1:N}(\mathcal{S}, \mathcal{A}, \boldsymbol{P}, \boldsymbol{r}_{1:N}, \gamma)$ is a factored MDP if each agent $i$ has neighbor set $Z_i \in [N]$ such that its transition is affected by all its neighbors, i.e. $P(s'_i \mid \boldsymbol{s}, \boldsymbol{a}) = P(s'_i \mid s_{Z_i}, a_{Z_i})$.*

The neighbor set $|Z_i|$ is often assumed to be much smaller compared to the number of agents $N$. This helps to encode exponentially large system very compactly. We show this idea can also be captured in Markov entanglement. Consider the measure of Markov entanglement w.r.t ATV distance in Eq. (7),

$$
\begin{aligned}
E_A(\boldsymbol{P}_{AB}^\pi) &= \min_{\boldsymbol{P}_A} \max_{(\boldsymbol{s}, \boldsymbol{a}) \in \mathcal{S} \times \mathcal{A}} D_{\mathrm{TV}} \left( \boldsymbol{P}_{AB}^\pi(\cdot, \cdot \mid \boldsymbol{s}, \boldsymbol{a}), \boldsymbol{P}_A(\cdot, \cdot \mid s_A, a_A) \right) \\
&= \min_{\boldsymbol{P}_A} \max_{(\boldsymbol{s}, \boldsymbol{a}) \in \mathcal{S} \times \mathcal{A}} D_{\mathrm{TV}} \left( \boldsymbol{P}_{AB}^\pi(\cdot, \cdot \mid s_{Z_A}, a_{Z_A}), \boldsymbol{P}_A(\cdot, \cdot \mid s_A, a_A) \right).
\end{aligned}
$$

Thus we conclude the agent-wise Markov entanglement will only depend on its neighbor set.

---

**Meta Algorithm 2:** Q-value Decomposition with Shared Reward

---

**Require:** Global policy $\pi$; horizon length $T$.

1: Execute $\pi$ for $T$ epochs and obtain $\mathcal{D} = \left\{ (s_{AB}^t, a_{AB}^t, r_{AB}^t, s_{AB}^{t+1}, a_{AB}^{t+1}) \right\}_{t=1}^{T-1}$.

2: Each agent $i \in \{A, B\}$ fits $Q_i^\pi$ using local observations $\mathcal{D}_i = \left\{ (s_i^t, a_i^t, r_i, s_i^{t+1}, a_i^{t+1}) \right\}_{t=1}^{T-1}$ where the local reward $(\boldsymbol{r}_A, \boldsymbol{r}_B)$ is learned via solving

$$\min_{\boldsymbol{r}_A, \boldsymbol{r}_B} \sum_{t=1}^{T} \left( r_{AB}^t(\boldsymbol{s}, \boldsymbol{a}) - (r_A(s_A^t, a_A^t) + r_B(s_B^t, a_B^t)) \right)^2.$$

---

## J.4 Fully cooperative Markov games

In fully cooperative settings, only a global reward will be reviewed to all agents. Unlike the modeling in section 2, this global reward may not necessarily be decomposed as the summation of local rewards. In this case, we propose meta algorithm 2 as an extension of meta algorithm 1.

This algorithm follows similar framework of meta algorithm 1 and differs at we now learn the closet local reward decomposition from data. When the reward is completely decomposable, meta algorithm 2 recovers meta algorithm 1. Thus intuitively, the more accurate we can decompose the global reward, the less decomposition error we have. Formally, we define the measure of reward entanglement

$$e(\boldsymbol{r}_{AB}) := \min_{\boldsymbol{r}_A, \boldsymbol{r}_B} \| \boldsymbol{r}_{AB} - (\boldsymbol{r}_A \otimes \boldsymbol{e} + \boldsymbol{e} \otimes \boldsymbol{r}_B) \|_{\mu_{AB}^\pi}. \tag{17}$$

This measure characterizes how accurate we can decompose the global reward under stationary distribution. We then obtain an extension of Theorem 4

**Proposition 13.** *Consider a fully cooperative two-agent Markov system $\mathcal{M}_{AB}$. Given any policy $\pi \colon \mathcal{S} \to \Delta(\mathcal{A})$ with the measure of Markov entanglement $E_A(\boldsymbol{P}_{AB}^\pi), E_B(\boldsymbol{P}_{AB}^\pi)$ w.r.t the $\mu_{AB}^\pi$-weighted agent-wise total variation distance and the measure of reward entanglement $e(\boldsymbol{r}_{AB})$, it holds*

$$\left\| Q_{AB}^\pi - (Q_A^\pi \otimes \boldsymbol{e} + \boldsymbol{e} \otimes Q_B^\pi) \right\|_{\mu_{AB}^\pi} \leq \frac{e(\boldsymbol{r}_{AB})}{1 - \gamma} + \frac{4\gamma \left( E_A(\boldsymbol{P}_{AB}^\pi) r_{\max}^A + E_B(\boldsymbol{P}_{AB}^\pi) r_{\max}^B \right)}{(1 - \gamma)^2},$$

*where $r_{\max}^A, r_{\max}^B$ is the bound of optimal solution of Eq. (17).*

Although Proposition 1 offers a theoretical guarantee for general two-agent fully cooperative Markov games, its utility is greatest in systems with low reward and transition entanglement. Fully cooperative settings remain inherently challenging–for instance, even the asymptotically optimal Whittle Index may achieve only a $\frac{1}{N}$-approximation ratio for RMABs with global rewards [34]. In practice, most research [38, 35] relies on sophisticated deep neural networks to learn decompositions in such settings. We thus defer a more refined analysis of fully cooperative scenarios to future work.

## K Simulation environments

In this section, we empirically study the value decomposition for index policies. Our simulations build on a circulant RMAB benchmark, which is widely used in the literature [3, 52, 10, 18].

**Circulant RMAB**  A circulant RMAB has four states indexed by $\{0, 1, 2, 3\}$. Transition kernels $P_a = p(s, 0, s')_{s, s' \in S}$ for action $a = 0$ and $a = 1$ are given by

$$\boldsymbol{P}_0 = \begin{pmatrix} 1/2 & 0 & 0 & 1/2 \\ 1/2 & 1/2 & 0 & 0 \\ 0 & 1/2 & 1/2 & 0 \\ 0 & 0 & 1/2 & 1/2 \end{pmatrix}, \quad \boldsymbol{P}_1 = \begin{pmatrix} 1/2 & 1/2 & 0 & 0 \\ 0 & 1/2 & 1/2 & 0 \\ 0 & 0 & 1/2 & 1/2 \\ 1/2 & 0 & 0 & 1/2 \end{pmatrix}.$$

The reward solely depends on the state and is unaffected by the action:

$$r(0, a) = -1,\ r(1, a) = 0,\ r(2, a) = 0,\ r(3, a) = 1; \forall a \in \{0, 1\}.$$

We set the discount factor to $\gamma = 0.5$ and require $N/5$ arms to be pulled per period. Initially, there are $N/6$ arms in state 0, $N/3$ arms in state 1 and $N/2$ arms in state 2, the same as [52]. We then test an index policy with priority: state $2 >$ state $1 >$ state $0 >$ state 3.

## K.1 Monte-Carlo estimation of Markov entanglement

For each RMAB instance, we simulate a trajectory of length $T = 6N$ and collect data for the later $5N$ epochs. Notice RMAB is a special instance of WCMDP, we thus apply the result in Proposition 9

$$E_i(\boldsymbol{P}_{1:N}^\pi) \le \frac{1}{2} \min_{\pi'} \sum_{\boldsymbol{s}} \mu_{1:N}^\pi(\boldsymbol{s}) \sum_{a_i} |\pi(a_i \mid \boldsymbol{s}) - \pi'(a_i \mid s_i)|$$

$$\approx \frac{1}{2} \min_{\pi'} \frac{1}{T} \sum_{t=1}^{T} \sum_{a_i} |\pi(a_i \mid \boldsymbol{s}) - \pi'(a_i \mid s_i)| \tag{18}$$

Notice Eq. (18) is *convex* for $\pi'$ and $\pi'$ only takes support of size $|S||A| = 8$. we thus apply efficient convex optimization solvers. We replicate this experiment for 10 independent runs to obtain the mean estimation and standard error in the left panel of Figure 1.

## K.2 Learning local Q-values

For each RMAB instance, we simulate a trajectory of length $T = 6N$, reserving the later $T = 5N$ epochs as the training phase for each agent to fit local Q-value functions. During testing, we estimate the $\mu$-weighted decomposition error using 50 simulations sampled from the stationary distribution.

The ground-truth $Q_{1:N}^\pi$ is approximated via Monte Carlo learning [39], with each estimate derived from 30-step simulations averaged over $3N$ independent runs. Due to the high computational cost of Monte Carlo methods—especially for very large RMABs—we limit the training phase to 10 independent runs and use the mean local Q-value as an approximation. Error bars represent the standard error for both Monte Carlo estimates and $\mu$-weighted decomposition errors.

In addition to $\mu$-weighted error, we also introduce a concept of relative error, defined as $\left\| Q_{1:N}^\pi(\boldsymbol{s}, \boldsymbol{a}) - \sum_{i=1}^{N} Q_i^\pi(s_i, a_i) \right\|_{\mu_{1:N}^\pi} / \|Q_{1:N}^\pi\|_{\mu_{1:N}^\pi}$. This relative error reflects the approximate ratio of our value decomposition. We present our simulation results below.

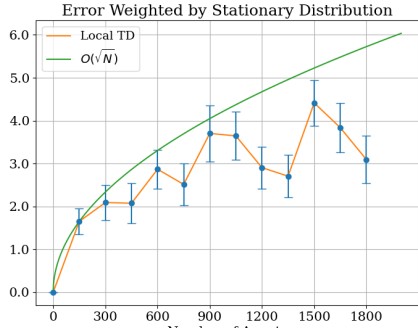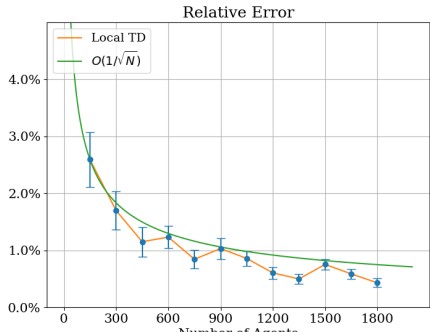

Figure 2: Value Decomposition error in circulant RMAB under an index policy. *Left:* $\mu$-weighted decomposition error. *Right:* Relative error, $\|\text{decomposition error}\|_\mu / \|Q_{1:N}^\pi\|_\mu$

It immediately follows that the $\mu$-weighted error grows at a sublinear rate $\mathcal{O}(\sqrt{N})$ and the relative error decays at rate $\mathcal{O}(1/\sqrt{N})$. This justifies our theoretical guarantees in Theorem 6. Furthermore, we notice the relative error is no larger than 3% over all data points. As a result, the meta algorithm 1 is able to provide a very close approximation especially for large-scale MDPs even with small amount of training data $T = 5N$ while the global state space has size $|S|^N$.

## K.3 Sample Complexity and Computation

While each RMAB instance has an exponentially large state space $|S|^N$, we show that our empirical estimation of Markov entanglement—along with the decomposition error—converges quickly. Specifically, we illustrate these errors for an RMAB instance with with 900 agents in Figure 3. As exhibits in Figure 3, both errors decay and converges within $T = 3N$ samples. Furthermore,

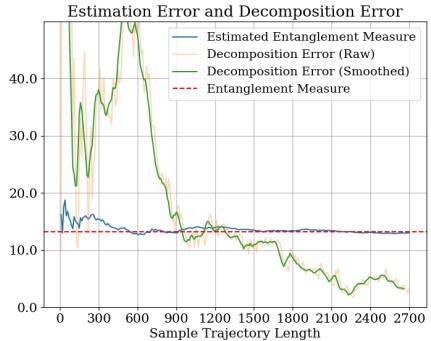

Figure 3: Different errors in RMAB with 900 agents: empirical estimation of Markov entanglement (blue); $\mu_{1:N}^{\pi}$-weighted decomposition error (green); the true measure of Markov estimated with $T = 10N$ samples (red dashed line).

the empirical estimation of Markov entanglement converges in $T < N$ samples, demonstrating its efficiency. Finally, we use standard convex optimization solvers to compute Markov entanglement, which can be run efficiently on a single CPU.

