# OpenReview forum: "Multi-agent Markov Entanglement"
_NeurIPS.cc/2025/Conference — NeurIPS 2025 spotlight_

### Official Review · Reviewer_bQ7W · 2025-07-02

**Clarity:** 3
**Significance:** 3
**Originality:** 3
**Rating:** 5
**Confidence:** 3

**Summary:**

This paper shows that the concept of entanglement (analogous to the idea of quantum entanglement) can be used to mathematically justify the why value decomposition works in reinforcement learning.

**Questions:**

- Can you provide a clearer discussion on how your concept truly connects to quantum entanglement?
- The analysis will only be valuable if you can scale it to more than two agents.
- Can you expand the experimental results to more varied use cases?
- Do your results still hold under non-stationarity?

**Ethical Concerns:**

["NO or VERY MINOR ethics concerns only"]

**Final Justification:**

I maintain my review and I see this paper as a useful contribution. The authors should, however, revise the work to omit the analogy to quantum as it does not really stand and could mislead the community.

**Limitations:**

Limitations are adequate

**Quality:**

3

**Strengths And Weaknesses:**

Strengths:
+ The theoretical results are very rigorous and the insights are important.
+ The link between entanglement and value decomposition is novel and this is an important early work in this space.
+ The two agent insights are useful.

Weaknesses:
- The experimental results are very limited.
- Scalability with the number of agents is not discussed.
- The linkage to quantum entanglement is rather tenuous.

---

> ### Author Rebuttal · Authors · 2025-07-31
>
> Thank you for the helpful review! In response to your questions:
>
> # More Experimental Results
> We have conducted additional experiments to investigate how the performance of a value decomposition based algorithm [1] varies with different levels of system entanglement. Please see our response to Reviewer H55K for complete experimental details and results.
>
> # Generalize to Multi-agent
> Our framework naturally extends to multi-agent scenarios, as briefly introduced in Section 5 and comprehensively discussed in Appendix H. All concepts and results established for two-agent MDPs maintain their validity in the multi-agent setting. Specifically, we attach Theorem 7 (line 688) in Appendix H here for reference
>
> ### Theorem 7
> Consider a $N$-agent MDP and policy $\pi$ with the measure of Markov entanglement $E_i(P_{1:N})$ w.r.t the $\mu^\pi_{1:N}$-weighted agent-wise total variation distance, it holds the decomposition error in $\mu^\pi_{1:N}$-norm is bounded by the measure of Markov entanglement,
> $$\\| Q^\pi_{1:N}(s_{1:N},a_{1:N})-\sum_{i=1}^N Q^\pi_i(s_i,a_i) \\| \leq \frac{4\gamma (\sum_{i=1}^NE_i(P_{1:N})r_{\max}^i)}{(1-\gamma)^2}\\,.$$
>
> # Relations to Quantum Entanglement
> So far, the connection is primarily conceptual: we observe a mathematical structure in multi-agent Markov systems that parallels quantum entanglement, and our definition of the measure is inspired by its quantum counterpart. We also believe that additional concepts from quantum physics, such as information scrambling—the process by which initially localized information becomes delocalized throughout a quantum system—may have meaningful analogues in the multi-agent Markov context. Furthermore, this shared mathematical structure suggests that quantum computers could potentially accelerate computations in multi-agent reinforcement learning, helping to overcome the curse of dimensionality (it will be exciting if this speedup is exponential over the classical computers). The graphical frameworks such as 1D/2D chains developed for understanding entanglement (studied in quantum simulation through a lens of tensor networks) may also find novel applications in the Markov setting. Overall, we believe this work opens a door between two distinct fields, and we are excited to see how advances in quantum theory might benefit multi-agent RL—and perhaps vice versa.
>
> # Extension to Non-stationary MDPs
> Good question! We believe our results can be extended to finite-time and non-stationary MDPs. We note that to achieve an entry-wise error bound, our proof does not require the transition data to be sampled from the stationary distribution; it only requires that the sampling distribution has positive measure for all state-action pairs. We leave further extensions to future work.
>
>
> [1] Sunehag, Peter, et al. "Value-decomposition networks for cooperative multi-agent learning." arXiv preprint arXiv:1706.05296 (2017).

---

> ### Comment · Reviewer_bQ7W · 2025-08-01
>
> I thank the authors for the response. While I remain positive of the work, I think the connection to quantum entanglement is tenuous and somewhat misleading  to the community,. I would maintain my score only if the authors would thoroughly revise the work to make the discussion on entanglement rather limited and down to earth. The current version of the paper somewhat makes that connection a central contribution but it isn’t even a one-to-one analogy. I would also revise the title and omit entanglement from it.

---

> > ### Author Response · Authors · 2025-08-02
> >
> > We thank the reviewer for the valuable comments and suggestions! The mathematical form of quantum entanglement indeed initially inspired our study; however, we agree that this connection is limited and that an MDP system does not involve any underlying physics of quantum entanglement. We view this connection as a mathematical inspiration rather than a core contribution. Following the reviewer’s advice, we will revise the paper to clarify this point, make the discussion down-to-earth, and ensure readers are not misled into expecting any physical interpretation related to quantum entanglement.
> >
> > Meanwhile, as suggested by Reviewer H55K, we will include a discussion on influence-based multi-agent RL to provide an alternative intuition supporting our value decomposition analysis.
> >
> > Additionally, we will reintegrate our N-agent results from Appendix H into the main paper in the final version. The two-agent scenario was used solely for educational and illustrative purposes.
> >
> > We hope these revisions address the reviewer’s comments and welcome any further feedback.

---

### Official Review · Reviewer_H55K · 2025-07-02

**Clarity:** 4
**Significance:** 3
**Originality:** 3
**Rating:** 5
**Confidence:** 3

**Summary:**

This paper provides a theoretical analysis of the error introduced by approximating a joint value function (across multiple agents) as linear combination of individual value functions, referred to as value decomposition. In particular, they introduce a measure of Markov entanglement, which bounds the error of such decompositions. To estimate this entanglement, they propose to measure the total variation distance between the transition matrix of focal agent $A$’s local state (and its next action/policy) conditioned only on its own current local state and action, and the transition matrix and policy conditioned on the global state and actions.

**Questions:**

1. The proposed method to estimate Markov entanglement seems to be closely related to common counterfactual methods in MARL used to estimate influence. For example, [1] uses a similar measure of information-theoretic influence to encourage agents to explore highly entangled transitions $p^{\boldsymbol{\pi}}\left(\boldsymbol{s}, \boldsymbol{a}, s_2^{\prime}\right)\left[\log p^{\boldsymbol{\pi}}\left(s_2^{\prime} \mid \boldsymbol{s}, \boldsymbol{a}\right)-\log p^{\boldsymbol{\pi}}\left(s_2^{\prime} \mid s_2, a_2\right)\right]$, and [2] proposes a similar mutual information-based method for entanglement of policies via the causal influence of one agent’s actions on another agent’s action. While this work uses it to bound value decomposition error rather than as intrinsic rewards for exploration, I would be interested to hear the authors discuss how their notion of Markov entanglement relates to these previous works.
2. In J.4, Proposition 12 seems to be circular in some sense — bounding the error of value decomposition in terms of reward entanglement, which itself measures the accuracy of a global reward decomposition over the stationary state distribution. Can the authors clarify the contribution of this point?
3. Apart from index policies, I would like to see more experiments to demonstrate the practical ramifications of using value decomposition in highly entangled MDPs with function approximation, e.g., a toy environment where the transition matrix/rewards can be altered to tune the level of entanglement, showing worse asymptotic performance with increasing entanglement for common decomposition methods such as VDNs [3] or QMIX [4]. I acknowledge that this would require a fair bit of engineering work, but might be useful for a camera-ready copy to make this more relevant to a wider body of MARL practitioners — I find that experiments only on index policies provide a rather narrow scope to demonstrate the utility of this approach.

[1] Wang, T., Wang, J., Wu, Y. & Zhang, C. Influence-Based Multi-Agent Exploration. in *Eighth International Conference on Learning Representations* (2020).

[2] Jaques, N. *et al.* Social Influence as Intrinsic Motivation for Multi-Agent Deep Reinforcement Learning. in *Proceedings of the 36th International Conference on Machine Learning* 3040–3049 (PMLR, 2019).

[3] Sunehag, P. *et al.* Value-Decomposition Networks For Cooperative Multi-Agent Learning.

[4] Rashid, T. *et al.* Monotonic value function factorisation for deep multi-agent reinforcement learning. *J. Mach. Learn. Res.* **21**, 178:7234-178:7284 (2020).

**Ethical Concerns:**

["NO or VERY MINOR ethics concerns only"]

**Final Justification:**

The Inventory Simulator experiments clearly demonstrate the relationship between increased entanglement and the performance degradation of value decomposition-based methods, which addressed my concerns about the actual effects of Markov entanglement. They also addressed my questions about J.4.

As said in my response, while I believe that an alternative framing and experiments on more commonly used MARL environments would increase the impact of this work, I believe that it provides a valuable contribution to the community and a solid theoretical framework to investigate decomposition-based approximations. Therefore, I will raise my score to a 5 - with limited experimental evaluation being the primary flaw that precludes a 6 (however, I do not think this merits a 4 as this paper's contribution is primarily theoretical).

**Limitations:**

Yes

**Quality:**

3

**Strengths And Weaknesses:**

**Strengths**

1. The idea of entanglement makes intuitive sense — it seems natural that value functions can be linearly decomposed into their component parts if their transition distributions are independent.
2. The motivation for the work is clearly laid out, and I was able to follow the argument clearly throughout despite not being familiar with some of the experiment settings and index policies.

**Weaknesses**

1. The empirical evaluation is limited, being only a single demonstration for index policies that demonstrates sublinear decomposition error scaling. However, I would argue that this is not a significant weakness given the primarily theoretical nature of the paper.
2. I believe that the paper’s notion of Markov entanglement is missing some key relevant work (expanded further upon in Questions).
3. Small nit: typo in Meta Algorithm 1 “Leaning”

---

> ### Author Rebuttal · Authors · 2025-07-31
>
> Thanks a lot for your review! Your suggestions are very insightful in helping us improve the paper and build strronger connections to the MARL literature.
>
> # New Empirical Evaluations
> We have conducted additional empirical evaluations to examine how VDN's performance varies with different levels of system entanglement. These experiments utilize a synthetic inventory control environment, which we describe in detail below.
>
> ## An Inventory Simulator
> A single-product inventory system consists of $k$ warehouses and 1 central factory. We denote the inventory level at each warehouse as $n_{1:k}$ and at factory as $n_f$. At each timestep,
> - Each warehouse $i\in[k]$ observes an customer order of $o_i$ products. $o_i$ is drawn from $\\{\underline{O_i},\overline{O_i}\\}$ according to a Bernoulli distribution with parameter $p_i$.
> - Central factory's action is to decide how many new products to produce (an integer $a_f \leq A_f$).
> - Each Warehouse's action is to decide how many products to order from the central factory (an integer $a_i \leq A_i$)..
> - Allow backlogged order. An order not fulfilled this timestep can be fulfilled at future timesteps.
> - Local state of each warehouse is $(n_i, o_i)$, the current inventory level and the current order. The central factory has state $(n_f)$ as its inventory.
> - The transition for each warehouse follows $$n_{i}^{t+1}=n_{i}^t-o_{i}^t+a_{i}^t.$$Similarly, for the central factory, it transits as$$n_{f}^{t+1}=n_f^t+a_f^t-\sum_{i=1}^ka_{i}^t.$$
> - Reward for local warehouse $r(n_i,o_i,a_i)=h_i\min\\{n_i,0\\}-c_i\max\\{n_i,0\\}$ where $h_i$ is the pentalty for the backlogged order and $c_i$ for the cost of storage. The reward for central factory is $r(n_f, a_f)=h_f\min\\{n_f,0\\}-c_f\max\\{n_f,0\\}$ where $h_f,c_f$ is the corresponding penalty.
>
> Note that the central factory's state transition depends not only on its local action but also on the actions of all warehouses. This interdependence creates potential system entanglement. Intuitively, entanglement increases with the number of warehouses. We therefore control the system's entanglement by adjusting $k$, the number of warehouses.
>
> ## Benchmark
> We implement VDN in [1] as the benchmark algorithm and test its performance under different levels of entanglement. More implementation details will be added in the next version of the paper. The optimal policy has access to the global information, thus it can fully avoid the cost. In this case, the factory supplies exactly the known order amount. Consequently, the optimal value function achieves zero cost. We then summarize the empirical results as follows
>
>
> |       | Estimated Entanglement | Global Optimal | VDN    |
> | ----- | ---------------------- | -------------- | ------ |
> | $k=1$ | 0.50                   | 0              | -9.72  |
> | $k=3$ | 0.81                   | 0              | -25.28 |
> | $k=5$ | 0.92                   | 0              | -36.94 |
> | $k=9$ | 0.98                   | 0              | -60.20 |
> |||||
>
> We have the following observation
> - VDN's performance degrades as system entanglement increases. This aligns with VDN's reliance on local value functions for action selection: as entanglement grows, local state information becomes increasingly insufficient for optimal decision-making.
>
> However, we also note that entanglement may not always determine VDN's performance. Through its centralized training framework, VDN can encode global reward/state information in its local value functions. As shown in our Section 3.1 and Appendix E, certain entangled MDPs still permit exact decomposition when local values effectively capture global reward information.
>
>
> # Relations to Influenced-based MARL
> Great question! It turns out the mutual information can be viewed as the measure of Markov entanglement under KL-divergence. Specifically, we can rewrite mutual information in [2] as $$\begin{aligned}&I(S_2^\prime, A_2^\prime; S_2, A_2| S_1,A_1)\\\\=&\sum_{s_{1:2},a_{1:2},s_2^\prime,a_2^\prime}p^\pi(s_{1:2},a_{1:2},s_2^\prime,a_2^\prime)[\log p^\pi(s_2^\prime,a^\prime_2|s_{1:2},a_{1:2})-\log p^\pi(s_2^\prime,a_2^\prime|s_{2},a_{2})]\\\\
>  =&\sum_{s_{1:2},a_{1:2}}\mu^\pi(s_{1:2},a_{1:2}) D_{KL}(p^\pi(\cdot,\cdot|s_{1:2},a_{1:2}) || p^\pi(\cdot,\cdot|s_{2},a_{2}) ) \\,. \end{aligned}$$
> This is highly related to our measure of Markov entanglement under a $\mu^\pi$-weighted agent-wise KL-divergence, which we can define as
> $$E_2(P_{1:2})=\min_{P_2} \sum_{s_{1:2},a_{1:2}}\mu^\pi(s_{1:2},a_{1:2}) D_{KL}(p^\pi(\cdot,\cdot|s_{1:2},a_{1:2}) || P_2(\cdot,\cdot|s_{2},a_{2}) )$$
> Intuitively, the measure of Markov entanglement can be viewed as how closely one agent can be approximated as an independent subsystem. This characterization aligns naturally with mutual information. Furthermore, since KL-divergence provides an upper bound for total variation distance, it consequently bounds our Markov entanglement measure relative to the ATV distance introduced in our paper. This connection demonstrates that influence-based MARL methods naturally fit within our theoretical framework, corresponding to a specialized distance measure.
>
> # Reward Entanglement in Appendix J.4
> This section extends our analysis to scenarios where the global reward cannot be decomposed. In such cases, we propose learning an approximate local decomposition of the global reward, which is then used to derive local value functions. Our theoretical results demonstrate that when the global reward admits a good approximation via summed local rewards (under the stationary distribution), the decomposition error remains bounded (under the stationary distribution).
>
>
>
>
>
>
>
> [1] Sunehag, Peter, et al. "Value-decomposition networks for cooperative multi-agent learning." arXiv preprint arXiv:1706.05296 (2017).
>
> [2] Wang, Tonghan, et al. "Influence-based multi-agent exploration." arXiv preprint arXiv:1910.05512 (2019).

---

> > ### Comment · Reviewer_H55K · 2025-08-01
> >
> > I appreciate the additional Inventory Simulator experiments, which clearly demonstrate the relationship between increased entanglement and the performance degradation of value decomposition-based methods, and for their clarifying comments on the proof in Appendix J.4.
> >
> > I also thank the authors for their thoughtful responses relating their work to influence-based MARL. I am inclined to agree with Reviewer bQ7W in the sense that the notion of Markov entanglement can be made more intuitive to the RL community from the lens of mutual information (MI) between agent actions, rather than quantum entanglement, given that the formulations very closely related. However, I also recognize that inspiration can come from many avenues and certainly would not fault this work for adopting a physics-based approach, as is common in many other areas of ML.
> >
> > While I believe that an alternative framing and experiments on more commonly used MARL environments would increase the impact of this work, I believe that it provides a valuable contribution to the community and a solid theoretical framework to investigate decomposition-based approximations. Accordingly, **I will raise my score to a 5 (accept)**, and ask that the authors consider including the above discussion of related MI approaches in a camera-ready copy.

---

> > > ### Author Response · Authors · 2025-08-02
> > > **Thank you!**
> > >
> > > We thank the reviewer for the insightful and sharp comments. The connection between mutual information in multi-agent RL and our study is indeed thought-provoking!
> > >
> > > As suggested, we will add a more thorough discussion of mutual information approaches in the final version and revise the framing where appropriate. We also plan to incorporate additional commonly used multi-agent RL experiments. These comments are extremely helpful for improving our paper, and we sincerely appreciate them.

---

### Official Review · Reviewer_tNC3 · 2025-07-02

**Clarity:** 3
**Significance:** 2
**Originality:** 3
**Rating:** 5
**Confidence:** 2

**Summary:**

This paper establishes a rigorous theoretical foundation for value decomposition in multi-agent reinforcement learning. The authors introduce Markov entanglement—an analogue of quantum entanglement—to characterize when a joint transition probability is separable and thus admits exact value decomposition. They prove that exact decomposition holds if and only if the Markov entanglement is zero, and they derive upper bounds on the approximation error in terms of an agent-wise total variation entanglement measure.

As an application, they show that under standard index policies for Restless Multi-Armed Bandits, the Markov entanglement scales as $O(1/\sqrt{N})$, and they confirm this rate via numerical experiments on circulant RMAB benchmarks. Additionally, they propose a simple Monte Carlo–based convex optimization procedure to efficiently estimate the entanglement, allowing practitioners to verify the feasibility of value decomposition without ever computing the full global Q-function.

**Questions:**

**Q1.**

Unlike quantum entanglement, Markov entanglement allows the coefficients $x_j$ to be negative. What is the fundamental reason behind this difference? Could you elaborate on why the constraint $x_j \ge 0$ is not imposed in the definition of Markov entanglement?


**Q2.**

I’m curious about the motivation for choosing the restless $N$-armed bandit setting for the experiments. Why was this particular environment selected?

**Ethical Concerns:**

["NO or VERY MINOR ethics concerns only"]

**Final Justification:**

The paper presents a novel theoretical foundation, well supported by a range of experiments. The authors have addressed my questions appropriately, and I have also benefited from discussions with other reviewers. Overall, I believe the paper has sufficient academic merit to warrant a final rating of 5.

**Limitations:**

yes

**Paper Formatting Concerns:**

I did not find any significant formatting issues in the paper.

**Quality:**

2

**Strengths And Weaknesses:**

**Strengths**

The paper establishes a rigorous theoretical foundation for value decomposition in MARL.
It provides a clean characterization of when value decomposition is possible, via the novel and well-defined concept of Markov entanglement.
The authors derive explicit upper bounds on the decomposition error in terms of an agent-wise TV entanglement measure, covering both worst-case and stationary distribution weighted scenarios.
While the main text focuses on two agents for clarity, the framework is extended to the general $N$-agent settings in the appendix.
The paper includes appropriate numerical results using the restless multi-armed bandit benchmark, supporting the theory with empirical evidence.

Overall, the paper is organized and well written.

**Weaknesses**

**W1.**

While the theory is well-developed, its practical relevance to more complex MARL domains remains unclear, as the evaluation is limited to a syntheic circulant RMAB benchmark.

**W2.**

I think assuming the decomposition $r_{AB}(s_A,s_B,a_A,a_B)=r_A(s_A,a_A)+r_B(s_B,a_B)$ is restrictive when agents compete or cooperate to achieve a goal. In such case, the interaction should be modeled differently. I believe most multi-agent scenarios fall into this category, where reward decomposition is unnatural.

---

> ### Author Rebuttal · Authors · 2025-07-31
>
> Thank you for the helpful review! In response to your questions:
>
> # More Evaluations
> We have conducted additional experiments to investigate how the performance of a value decomposition based algorithm [1] varies with different levels of system entanglement. Please see our response to Reviewer H55K for complete experimental details and results
>
> # Decomposition of Reward
> We argue that many practical MARL settings fall into our modeling, especially in the field of operations research (OR). Typical examples include
> - Ride-hailing system (e.g. [1][2]) The platform's overall revenue is the summation of the revenue of each driver.
> - Warehouse Robot routing (e.g. [3]) The local reward is the indicator of whether the agent completes a task. The global objective is the overall number of task completed.
> - Inventory Control (e.g. [4]) The overall operational cost is the summation of the costs of each warehouse.
> - The well-established class of Weakly-coupled MDPs (e.g. [5]) and RMABs (e.g. [6]) in OR literature. These models have been widely applied in practice, including in resource allocation, revenue management, healthcare, and many other fields.
>
> Furthermore, in Appendix J.2, we also provide a solution for scenarios where the global reward cannot be decomposed. We propose a concept called *reward entanglement* to characterize how well the global reward function can be approximated by a sum of local rewards. Building on this concept, we extend our analysis, demonstrating that the decomposition error can be bounded by considering both transition and reward entanglement.
>
> # Q1: Negative Coefficient in Markov Entanglement
> In Appendix C we thoroughly discuss this difference between Markov entanglement and quantum entanglement. In short, we did find a separable two-agent MDPs that can only be represented by linear combinations but not convex combinations of independent subsystems. This result justifies the necessity of negative coefficients $x_j <0$ and highlights a structural difference between Markov entanglement and quantum entanglement. Specifically, we construct the following transition matrix in $\mathbb R^{4\times 4}$ $$\left(\begin{array}{cccc}
> 0.5 & 0 & 0 & 0.5 \\\\
> 0.5 & 0 & 0 & 0.5 \\\\
> 0.5 & 0 & 0 & 0.5 \\\\
> 0 & 0.5 & 0.5 & 0
> \end{array}\right)\\,.$$ One can verify this transition matrix can not be represented by the convex combination of tensor products of transition matrices in $\mathbb R^{2\times 2}$.
>
>
>
>
> # Q2: RMAB as Evaluation Environment
> We choose RMAB out of the following reasons
> - RMAB forms the foundation for several award-winning industrial applications in operations research, including ride-hailing systems (DiDi, Lyft) [1,2] and Kenya's mobile healthcare platform [8]. We then adopt RMABs as our evaluation environment to empirically investigate why value decomposition based methods prove effective in large-scale systems and verify our theoretical justifications.
> - RMAB involves two major challenges in MARL
>    - *Scalability*: as demonstrated in our experiment, a RMAB instance usually contains thousands of agents. For example, the ride-hailing system in NYC contains thousands of active drivers.
>     -  *Action Constraint*: each agent's action is coupled with each other. For example, if one driver takes one order, other driver cannot fulfill the same order.
>
> These structures make RMAB an active research area in OR and MARL.
>
>
>
> [1] Qin, Z., et al. "Ride-hailing order dispatching at didi via reinforcement learning." INFORMS Journal on Applied Analytics 50.5 (2020): 272-286.
>
> [2] Azagirre, Xabi, et al. "A better match for drivers and riders: Reinforcement learning at lyft." INFORMS Journal on Applied Analytics 54.1 (2024): 71-83.
>
> [3] Chan, Shao-Hung, et al. "The league of robot runners competition: Goals, designs, and implementation." ICAPS 2024 System's Demonstration track. 2024.
>
> [4] Alvo, Matias, Daniel Russo, and Yash Kanoria. "Neural inventory control in networks via hindsight differentiable policy optimization." arXiv preprint arXiv:2306.11246 (2023).
>
> [5] Adelman, Daniel, and Adam J. Mersereau. "Relaxations of weakly coupled stochastic dynamic programs." Operations Research 56.3 (2008): 712-727.
>
> [6] Whittle, Peter. "Restless bandits: Activity allocation in a changing world." Journal of applied probability 25.A (1988): 287-298.
>
> [7] Sunehag, Peter, et al. "Value-decomposition networks for cooperative multi-agent learning." arXiv preprint arXiv:1706.05296 (2017).
>
> [8] Baek, Jackie, et al. "Policy optimization for personalized interventions in behavioral health." Manufacturing & Service Operations Management 27.3 (2025): 770-788.

---

> > ### Comment · Reviewer_tNC3 · 2025-08-03
> >
> > I appreciate the authors' detailed feedback. Their clarifications and additional experiments have resolved my concerns, so I am raising my rating to 5.

---

### Official Review · Reviewer_5Std · 2025-07-03

**Clarity:** 4
**Significance:** 3
**Originality:** 3
**Rating:** 5
**Confidence:** 3

**Summary:**

This paper investigates the equivalent condition of the linear-decomposability of the value function in multi-agent MDPs. The authors introduces the concept of Markov entanglement, inspired by the entanglement in quantum mechanics, to characterize property of the dynamics of the multi-agent MDP. They prove that the linear-decomposability of the value function is equivalent to the Markov disentanglement of the dynamics (in non-trivial cases). Beyond this results, they define a quantitative measure of Markov entanglement, and relate it to the value decomposition error. The paper further shows that index policies exhibit low entanglement and thus enjoy a $O(\sqrt N)$ scale of decomposition error. Finally, they show that the entanglement can be efficiently estimated, making it a useful tool for analyzing the linear decomposition error of the value function.

**Questions:**

See Weaknesses.

**Ethical Concerns:**

["NO or VERY MINOR ethics concerns only"]

**Final Justification:**

A technically solid paper. It proposes an interesting theory, bringing new insights into the understanding of value decomposition in the MARL field.

**Limitations:**

yes

**Quality:**

3

**Strengths And Weaknesses:**

### Strengths

- The paper is well-written and the presentation is clear.

- The paper provides a new perspective to understand linear decomposition of the value function in multi-agent MDPs.

- The theoretical results are comprehensive, and the paper also provides a method for estimating entanglement, making it applicable in practical settings.

### Weaknesses

- It would be better if the authors devote more discussion to the intuitive understanding of entanglement, and clarify how the proposed necessary and sufficient condition relates to other commonly used notions such as independence and Definition 3.1 in [1]. Discussion on the relationship between entanglement and coupling is also beneficial.

- The paper does not provide the efficiency analysis of the entanglement estimation method.

- (Minor) Linear decomposition of the value function is a fundamental problem in multi-agent RL, while it is not so effective in practice. Most of the existing empirical works focus on the monotonic decomposition[2]. It would be interesting to see the analysis of the monotonic decomposition.

[1] Dou, Zehao, Jakub Grudzien Kuba and Yaodong Yang. “Understanding Value Decomposition Algorithms in Deep Cooperative Multi-Agent Reinforcement Learning.” ArXiv abs/2202.04868 (2022): n. pag.

[2] Rashid, Tabish, Mikayel Samvelyan, C. S. D. Witt, Gregory Farquhar, Jakob N. Foerster and Shimon Whiteson. “QMIX: Monotonic Value Function Factorisation for Deep Multi-Agent Reinforcement Learning.” ArXiv abs/1803.11485 (2018): n. pag.

---

> ### Author Rebuttal · Authors · 2025-07-31
>
> Thank you for the helpful review! In response to your questions:
>
> # Intuition of Entanglement
> In the introduction, we highlight that Markov entanglement intuitively captures whether an MDP can be viewed as a *linear combination of independent subsystems*. To further clarify this concept, we provide an illustrative example in Section 3 (Line 191) comparing entanglement with coupling. Specifically, we construct an MDP that is coupled but *not* entangled, highlighting the fundamental distinction between the two notions.
>
> As a supplement, our example relies on a special MDP where two agents share the same randomness. This idea also has a direct parallel in quantum physics. Shared randomness plays a crucial role in characterizing the fundamental distinction between *classical correlation* and *quantum entanglement* in the quantum world, e.g. the famous Einstein-Podolsky-Rosen (EPR) paradox. When it comes to the Markov world, it delineates the difference between *coupling* and *Markov entanglement*. We plan to incorporate this perspective more explicitly in the next version of the paper.
>
> # Relations to Independence and Definition 3.1 in [1]
> Our notion of Markov entanglement strictly generalizes independence by allowing MDPs to be expressed as a *linear combination of independent subsystems*, allowing for coupled (non-independent) MDPs that are separable.
> Definition 3.1 in [1] also assumes the global reward as the summation of local rewards. It further requires the transition to decompose into a sum of specific functions. While the authors show their definition is equivalent not only to exact value decomposition but also to the exact decomposition under the Bellman operator, they admit that MDPs satisfying their definition are "rare in practice." Specifically, we find Definition 3.1 in [1] cannot accommodate certain independent cases.
>
>
> # Efficiency Analysis of the Entanglement Estimation
> We briefly present the efficiency results here, with a detailed analysis to be included in the next version of the paper. First, note that the optimization problem can be abstracted as follows: $\min_x F(x)=\min_{\\|x\\|=1} \mathbb E_y[\\|y-x\\|]=:L$ where $\\|\cdot\\|$ refers to the L1-norm. Denote one optimal solution $m=arg\min_x F(x)$. Similarly, let $L_n:=\min_{\\|x\\|=1} F_n(x)=\min_{\\|x\\|=1} \frac{1}{n}\sum_{i=1}^n\\|y_i-x\\|_1$ with one optimal solution $m_n=arg\min_x F_n(x)$. We summarize the results in the table below
>
> |            | Efficiency                                                       |
> | ---------- | ---------------------------------------------------------------- |
> | Asymptotic | $\sqrt n (L_n-L)\xrightarrow{d} \mathcal{N}(0,Var_y(\\\|y-m\\\|_1))$ |
> |     Non-asymptotic       |     $\| L_n-L \|\leq O(\sqrt\frac{\| S \|\| A \|}{n})$  where $\| S \|\| A \|$ is the size of local state-action pairs.  |
> |||
>
> We then briefly go through the proof sketch here
> - The asymptotic efficiency analysis is based on empirical process theory. Note that $L_n-L=F_n(m)-L+F_n(m_n)-F(m_n)+F(m_n)-F_n(m)$. For the first part, it's clear $\sqrt n(F_n(m)-L)\xrightarrow d \mathcal{N}(0,Var_y(\\|y-m\\|_1))$ following standard CLT. For the later part, one can verify $F_n(m_n)-F(m_n)\leq \sup_x |F_n(x)-F(x)|$ and $F(m_n)-F_n(m)\leq \sup_x |F_n(x)-F(x)|$. Thus it suffices to bound $\sup_x |F_n(x)-F(x)|$. We can verify the covering number of $F$ satisfies polynomial decay and thus apply standard analysis of uniform convergence (e.g. Theorem 37 in [2]). We have $\sup_x |F_n(x)-F(x)|=o_p(n^{-1/2})$. Finally, we derive the asymptotic result via Slutsky’s theorem. For simplicity, the proof above assumes that transitions are sampled i.i.d. from the stationary distribution. When the data is drawn from a non-episodic Markov chain, standard Markov chain CLT (e.g. [4]) results can be applied to obtain analogous conclusions.
> - The non-asymptotic analysis is based on generalization theory. We note that $|L_n-L|\leq R_n(F)+\sqrt{\frac{\log (1/\delta)}{n}}$ with probability at least $1-\delta$ and $R_n(F)$ is the empirical Rademacher complexity. We can bound $R_n(F)$ via the covering number and conclude its order $O(\sqrt\frac{|S||A|}{n})$.
>
> In summary, our entanglement estimation has efficiency that does not scale with the number of agents $N$.  In contrast, estimation of decomposition error without using Markov entanglement requires direct evaluation of global value functions, whose asymptotic efficiency has rate $\Theta(1/\mu_{\min})$ where $\mu_{\min}=\min \mu(s_{1:N},a_{1:N})$ (e.g. see [3]). In other words, the rate of direct estimation is $\Omega(|S|^N|A|^N)$. Thus the estimation of decomposition error via Markov entanglement is exponentially faster.
>
>
>
> # Beyond Linear Decomposition (Minor)
> We agree that linear value decomposition is a fundamental problem in MARL. We also acknowledge the rich body of work on non-linear decomposition methods (such as monotonic) that offer richer representations and enjoy greatempirical successes. Our work on linear decomposition and Markov entanglement can serve as a foundational framework for future analysis of such algorithms.
>
>
> [1] Dou, Zehao et.al, "Understanding Value Decomposition Algorithms in Deep Cooperative Multi-Agent Reinforcement Learning." ArXiv 2022.
>
> [2] Polldard, David, "Convegence of Stochastic Processes." Springer 1984
>
> [3] Chen, Shuze, et.al, "Experimenting on Markov Decision Processes with Local Treatments." Arxiv 2024
>
> [4] Galin L. Jones, "On the Markov chain central limit theorem." Arxiv 2004

---

> > ### Comment · Reviewer_5Std · 2025-07-31
> > **Reply**
> >
> > Thank you for your response. I have no further questions.

---

### Decision · Program_Chairs · 2025-09-17

**Decision:**

Accept (spotlight)

**Comment:**

Overall, the reviewers were quite favorable towards this paper. They had some questions which the authors easily addressed. I hope that the authors will review their manuscript in light of these questions and make edits aimed towards minimizing any ambiguities that might confuse future readers. In that vein, I hope that the authors will be careful to avoid leaning too heavily on the quantum entanglement motivation (per discussion with reviewer bQ7W) unless they can explain how the type of entanglement here is more like quantum entanglement than other cases of variable entanglement that are clearly understandable without resorting to quantum metaphors.